# Rupture-dependent breakdown energy in fault models with thermo-hydro-mechanical processes

Valère Lambert[1] and Nadia Lapusta[1,2]

[1]Seismological Laboratory, California Institute of Technology, Pasadena, CA 91125, USA
[2]Department of Mechanical and Civil Engineering, California Institute of Technology, Pasadena, CA 91125, USA

**Correspondence:** Valère Lambert (vlambert@caltech.edu)

**Abstract.** Substantial insight into earthquake source processes has resulted from considering frictional ruptures analogous to cohesive-zone shear cracks from fracture mechanics. This analogy holds for slip-weakening representations of fault friction that encapsulate the resistance to rupture propagation in the form of breakdown energy, analogous to fracture energy, prescribed in advance as if it were a material property of the fault interface. Here, we use numerical models of earthquake sequences with enhanced weakening due to thermal pressurization of pore fluids to show how accounting for thermo-hydro-mechanical processes during dynamic shear ruptures makes breakdown energy rupture-dependent. We find that local breakdown energy is neither a constant material property nor uniquely defined by the amount of slip attained during rupture, but depends on how that slip is achieved through the history of slip rate and dynamic stress changes during the rupture process. As a consequence, the frictional breakdown energy of the same location along the fault can vary significantly in different earthquake ruptures that pass through. These results suggest the need for re-examining the assumption of pre-determined frictional breakdown energy common in dynamic rupture modeling and for better understanding of the factors that control rupture dynamics in the presence of thermo-hydro-mechanical processes.

## 1 Introduction

Fault constitutive relations that describe the evolution of shear resistance with fault motion are critical ingredients of earthquake source modeling. When coupled with the elastodynamic equations of motion, these relations provide insight into the growth and ultimate arrest of ruptures. Earthquake source processes are often considered in the framework of dynamic fracture mechanics, where the earthquake rupture may be considered as a dynamically propagating shear crack or pulse (Ida, 1972; Palmer and Rice, 1973; Madariaga, 1976; Rice, 1980; Kostrov and Das, 1988; Heaton, 1990; Freund, 1990; Kanamori and Heaton, 2000; Rice, 2000; Kanamori and Brodsky, 2004; Rubin and Ampuero, 2005).

By analogy to cohesive-zone relations for Mode I opening cracks, slip-weakening laws have been commonly used to describe the dynamic decrease in shear resistance during sliding (Ida, 1972; Palmer and Rice, 1973; Madariaga, 1976; Kostrov and Das, 1988; Kanamori and Brodsky, 2004; Bouchon, 1997; Ide and Takeo, 1997; Olsen et al., 1997; Bouchon et al., 1998; Cruz-Atienza et al., 2009; Kaneko et al., 2017; Gallovic et al., 2019). Linear slip weakening is one of the simplest and most

commonly used versions, in which the shear resistance decreases linearly with slip from a peak of $\tau_{\text{peak}}$ to a constant dynamic

level $\tau_{\text{dyn}}$ achieved at slip of $D_c$ (Fig. 1).

  The breakdown energy $G$ is associated with the evolution of shear resistance from the initial shear stress $\tau_{\text{ini}}$ to the peak shear

resistance $\tau_{\text{peak}}$ and then breakdown to the minimum dynamic shear resistance $\tau_{\text{min}}$. It is a part of the overall energy partitioning

for dynamic ruptures, with the total strain energy change throughout the ruptured region ($\Delta W$) being separated into the radiated

energy $E_R$, breakdown energy $G$, and other residual dissipated energy (Kanamori and Rivera, 2006). The breakdown energy

is analogous to fracture energy from cohesive zone models of fracture mechanics (Palmer and Rice, 1973; Rice, 1980; Freund,

1990; Tinti et al., 2005) and hence it is thought to be relevant to rupture dynamics, e.g., rupture speed. For linear slip-weakening

friction, it is given by $G = (\tau_{\text{peak}} - \tau_{\text{dyn}})D_c/2$. The term "fracture energy," while initially associated with the creation of free

surfaces during tensile fracture, has been routinely used to refer broadly to inelastic dissipation relevant to the crack-tip motion

for both tensile and shear cracks, including contributions from off-fault damage creation, plastic work, and frictional heat (e.g.

Rice, 1980; Freund, 1990; Rice, 2006). However, here we follow the work of Tinti et al. (2005) in referring to this quantity

as the "breakdown" work, or energy, to further emphasize that $G$ can incorporate various physical sources of energy dissipation.

  More involved fault-constitutive laws are generally required to explain a number of aspects of faulting behavior, most no-

tably the restrengthening of faults in between earthquakes. Laboratory experiments have provided significant insight into the

rich behavior of shear resistance, with the frictional response at slip rates between $10^{-9} - 10^{-3}$ m/s being well-described by

rate-and-state friction laws (Dieterich, 2007). A number of previous studies have used models on rate-and-state faults to provide

insight into a number of earthquake and slow slip observations, such as sequences of earthquakes on an actual fault segment

and repeating earthquakes (Chen and Lapusta, 2009; Barbot et al., 2012; Dieterich, 2007, and references therein). While incor-

porating a more involved dependence of shear resistance on long-term healing, standard Dieterich-Ruina rate-and-state friction

has been shown to resemble linear slip weakening during dynamic rupture (Okubo, 1989; Cocco and Bizzarri, 2002; Lapusta

and Liu, 2009), providing further reinforcement of the notion that the breakdown of shear resistance during dynamic rupture

may be adequately described by linear slip-weakening behavior.


  Many studies have attempted to infer parameters of the slip-weakening shear resistance from the strong-motion data resulting

from natural earthquakes (Bouchon, 1997; Ide and Takeo, 1997; Olsen et al., 1997; Bouchon et al., 1998; Cruz-Atienza et al.,

2009; Kaneko et al., 2017; Gallovic et al., 2019). Such studies have noted substantial trade-offs in the inferred parameters

during such inversions, such as between the slip-weakening distance $D_c$ and strength excess $\tau_{\text{peak}} - \tau_{\text{ini}}$, where $\tau_{\text{ini}}$ is the initial

stress (Fig. 1). It has been presumed that the spatial distribution of static stress drop and breakdown energy may be the most

reliably determined features, as the stress drop can be inferred from the spatial distribution of slip and remaining variations

in rupture speed are largely controlled by the breakdown energy in such linear slip-weakening representations (Guatteri and

Spudich, 2000).

One of the most notable features of seismologically-inferred breakdown energies from natural earthquakes is that the average breakdown energy from the rupture process has been inferred to increase with the earthquake size (Abercrombie and Rice, 2005; Rice, 2006; Cocco and Tinti, 2008; Viesca and Garagash, 2015; Brantut and Viesca, 2017). Increase in breakdown energy with slip has also been observed in high-speed friction experiments (Nielsen et al., 2016; Selvadurai, 2019), although in some experiments the increase saturates after a given amount of weakening (Nielsen et al., 2016). Such findings are inconsistent with the breakdown energy being a fixed fault property as often assumed in linear slip-weakening laws and as approximately follows from standard rate-and-state friction with uniform characteristic slip-weakening distance (Perry et al., 2020), unless strong and very special heterogeneity is assumed in fault properties. For example, some modeling studies have assigned strongly heterogeneous $D_c$ and hence $G$ values to the fault, as if they are properties of the interface, with larger patches having significantly larger values of $D_c$ and hence $G$, and considered sequences of events over such interfaces (e.g. Ide and Aochi, 2005; Aochi and Ide, 2011).

Several theoretical and numerical studies have demonstrated that enhanced dynamic weakening, as widely observed at relatively high slip rates ($> 10^{-3}$ m/s) in laboratory experiments (Tullis, 2007; Di Toro et al., 2011), may explain the inferred increase in breakdown energy with slip (Rice, 2006; Viesca and Garagash, 2015; Brantut and Viesca, 2017; Perry et al., 2020). A number of different mechanisms have been proposed for such enhanced weakening, many of them due to shear heating. For example, thermal pressurization may occur due to the rapid shear heating of pore fluids during slip (Sibson, 1973; Andrews, 2002; Rice, 2006); if pore fluids are heated fast enough and not allowed to diffuse away, they pressurize and reduce the effective normal stress on the fault. Flash heating is another thermally-induced weakening mechanism, where the effective friction coefficient is rapidly reduced due to local melting of highly stressed micro-contacts along the fault (Rice, 1999; Goldsby and Tullis, 2011; Passelegue et al., 2014). Considerations of heat production during dynamic shear ruptures provide a substantial constraint for potential fault models, as field studies show no correlation between faulting and heat flow signatures and rarely suggest the presence of melt (Sibson, 1975; Lachenbruch and Sass, 1980). Models with enhanced weakening have been successful in producing fault operation at low overall prestress and low heat production (Rice, 2006; Noda et al., 2009; Lambert et al., in review) as supported by several observations (Brune et al., 1969; Zoback et al., 1987; Hickman and Zoback, 2004; Williams et al., 2004).

Numerical models have shown that the incorporation of thermally-activated enhanced weakening mechanisms during dynamic rupture can have profound effects on the evolution of individual ruptures, as well as the long-term behavior of fault segments, with the potential to make seemingly stable creeping regions fail violently during earthquakes (Noda and Lapusta, 2013), and for the deeper penetration of large ruptures, which may explain the seismic-quiescence of mature faults that have historically hosted large earthquakes (Jiang and Lapusta, 2016). Despite evolving dynamic resistance in such models, they can also be consistent with magnitude-invariant static stress drops (Perry et al., 2020).

At the same time, accounting for thermo-hydro-mechanical processes during dynamic rupture can clearly weaken or even remove the analogy between frictional shear ruptures and idealized shear cracks of fracture mechanics. The analogy is based on two key assumptions: 1) that the breakdown of shear resistance is concentrated in a small region near the rupture front, referred to as small-scale yielding, and 2) that there exists a constant residual stress level $\tau_{dyn} = \tau_{min}$ throughout the ruptured region during sliding (Palmer and Rice, 1973; Freund, 1990). For example, the relationship between rupture speed and fracture energy of linear elastic fracture mechanics is on valid under these assumptions. Clearly, these assumptions can become invalid when thermo-hydro-mechanical processes are considered. For example, shear heating can raise the pore fluid pressure in regions away from the rupture front and weaken the fault there, contributing to the breakdown of fault resistance away from the rupture tip and varying the dynamic resistance level. Furthermore, the shear heating itself would depend on the overall dissipated energy, making the fault weakening behavior, and hence "breakdown," depend on the absolute stress levels, and not just the stress changes, as typically considered by analogy with traditional fracture mechanics. Moreover, studies that infer dynamic parameters from natural earthquakes using dynamically-inspired kinematic models suggest more complicated evolutions of shear stress with slip, including heterogeneous dynamic resistance levels (Ide and Takeo, 1997; Bouchon et al., 1998; Tinti et al., 2005; Causse et al., 2013)

In this study, we use numerical models of earthquake sequences with enhanced weakening due to thermal pressurization to illustrate how the inclusion of thermo-hydro-mechanical processes during dynamic shear ruptures makes breakdown energy rupture-dependent, in that the value of both local and average breakdown energy vary among ruptures on the same fault, even with spatially uniform and time-independent constitutive properties. As such, the breakdown energy is not an intrinsic fault property, but develops different values at a given location, depending on the details of the rupture process, which in part depend on the prestress before the dynamic rupture achieved as a consequence of prior fault slip history. Moreover, the local breakdown energy is not uniquely defined by the amount of slip attained during rupture, but depends on how that slip was achieved through the complicated history of slip rate and dynamic stress changes throughout the rupture process. Additional fault characteristics that we do not consider here, such as heterogeneity in fault properties and dynamically-induced, evolving, inelastic off-fault damage (Dunham et al., 2011a, b; Roten et al., 2017; Withers et al., 2018) should result in qualitatively similar effects and add even more variability to the breakdown energy.

## 2    Description of numerical models

We conduct numerical simulations of spontaneous sequences of earthquakes and aseismic slip (SEAS) utilizing the spectral boundary integral method to solve the elastodynamic equations of motion coupled with friction boundary conditions, including the evolution of pore fluid pressure and temperature on the fault coupled with off-fault diffusion (Lapusta et al., 2000; Noda and Lapusta, 2010). Our simulations consider mode III slip on a 1-D fault embedded into a 2-D uniform, isotropic, elastic medium slowly loaded with a long-term slip rate $V_{pl}$ (Fig. 2). The simulations resolve the full slip behavior throughout earthquake sequences, including the nucleation process, the propagation of individual dynamic ruptures, as well as periods of post-seismic

and the interseismic slip between events that can last for months to hundreds of years.

Our fault models adopt the laboratory-derived Dieterich-Ruina rate-and-state friction law with the state evolution governed by the aging law (Dieterich, 1979; Ruina, 1983):

$$\tau = \overline{\sigma} f(V, \theta) = (\sigma - p) \left[ f_* + a \log \frac{V}{V_*} + b \log \frac{\theta V_*}{D_{\mathrm{RS}}} \right],$$ (1)

$$\dot{\theta} = 1 - \frac{V\theta}{D_{\mathrm{RS}}},$$ (2)

where $\overline{\sigma}$ is the effective normal stress, $\sigma$ is the normal stress, $p$ is the pore fluid pressure, $f_*$ is the reference steady-state friction coefficient at reference sliding rate $V_*$, $D_{\mathrm{RS}}$ is the characteristic slip distance, and $a$ and $b$ are the direct effect and evolution effect parameters, respectively. Other formulations for the evolution of the state variable exist, such as the slip law (Ruina, 1983) as well as various composite laws, and the formulation that best describes various laboratory experiments remains a topic of ongoing research (Bhattacharya et al., 2015, 2017; Shreedharan et al., 2019). However, the choice of the state evolution law should not substantially influence the results of this study, as the evolution of shear resistance during dynamic rupture within our simulations is dominated by the presence of enhanced weakening mechanisms. We use the version of the expressions (1) and (2) regularized for zero and negative slip rates (Noda and Lapusta, 2010).

During conditions of steady-state sliding ($\dot{\theta} = 0$), the friction coefficient is expressed as:

$$f_{ss}(V) = f_* + (a - b) \log \frac{V}{V_*}.$$ (3)

The combination of frictional properties $(a - b) > 0$ results in steady-state velocity-strengthening (VS) behavior, where stable slip is expected, and properties resulting in $(a - b) < 0$ lead to steady-state velocity-weakening (VW) behavior, where accelerating slip and hence stick-slip occur for sufficiently large regions (Rice and Ruina, 1983; Rice et al., 2001; Rubin and Ampuero, 2005).

An important, yet often underappreciated, implication of the rate- and state-dependent effects observed in laboratory experiments is that notions of static and dynamic friction coefficients, as well as the slip-weakening distance, are not well-defined and fixed quantities, as would be considered by standard linear slip-weakening laws (Cocco and Bizzarri, 2002; Rubin and Ampuero, 2005; Ampuero and Rubin, 2008; Lapusta and Liu, 2009; Barras et al., 2019; Perry et al., 2020). Instead, they depend on the history and current style of motion. For example, the dynamic friction, comparable to the steady-state friction at dynamic slip rates, depends on slip rate (Eq. 3), which can vary substantially throughout rupture and between different ruptures. Moreover, the peak friction and effective slip-weakening distance under standard rate-and-state friction depend on the history of motion through the state variable $\theta$, as well as the sliding rate during fast slip (Fig. 3). Let us consider a point with the same initial friction but different periods of inter-event healing, captured by increasingly larger values of the pre-rupture state variable. If the point is now driven to slide at a fixed sliding rate, the peak friction and slip-weakening distance would be larger for points that (i) have higher pre-rupture value of the state variable, representing better healed interfaces, and/or (ii) sliding at

faster slip rates (Fig. 3). For standard rate-and-state friction, these effects typically translate into generally mild variations in dynamic/static stress drop and breakdown energy, due to the logarithmic dependence of the shear stress evolution on slip rate, resulting in both static stress drop and breakdown energy being effectively rupture-independent (Cocco and Bizzarri, 2002; Rubin and Ampuero, 2005; Ampuero and Rubin, 2008; Lapusta and Liu, 2009; Perry et al., 2020), at least compared to the large variations of breakdown energy with slip inferred from natural earthquakes as discussed in the introduction. However, such variations in stress evolution become more substantial with enhanced dynamic weakening mechanisms that lead to stronger rate-dependent weakening.

Laboratory experiments indicate that the standard rate-and-state laws (Eqs. 1 - 2) provide good descriptions of frictional behavior at relatively slow slip rates ($10^{-9}$ to $10^{-3}$ m/s). However, at higher sliding rates, including average seismic slip rates of $\sim$1 m/s, additional enhanced weakening mechanisms can occur, such as the thermal pressurization of pore fluids. Thermal pressurization is governed in our simulations by the following coupled differential equations for the evolution of temperature and pore fluid pressure (Noda and Lapusta, 2010):

$$\frac{\partial T(y,z;t)}{\partial t} = \alpha_{\text{th}} \frac{\partial^2 T(y,z;t)}{\partial y^2} + \frac{\tau(z;t)V(z;t)}{\rho c} \frac{\exp(-y^2/2w^2)}{\sqrt{2\pi}w}, \tag{4}$$

$$\frac{\partial p(y,z;t)}{\partial t} = \alpha_{\text{hy}} \frac{\partial^2 p(y,z;t)}{\partial y^2} + \Lambda \frac{\partial T(y,z;t)}{\partial t}, \tag{5}$$

where $T$ is the pore fluid temperature, $\alpha_{\text{th}}$ is the thermal diffusivity, $\tau V$ is the shear heating source which is distributed over a Gaussian shear layer of half-width $w$, $\rho c$ is the specific heat, $y$ is the fault-normal distance, $\alpha_{\text{hy}}$ is the hydraulic diffusivity, and $\Lambda$ is the coupling coefficient that provides the change in pore pressure per unit temperature change under undrained conditions.

The total fault domain of size $\lambda$ is partitioned into a frictional region of size $\lambda_{\text{fr}}$ where we solve for the balance of shear stress and frictional resistance, as well as loading regions at the edges where the fault is prescribed to slip at a tectonic plate rate (Fig. 2A). The frictional interface is composed of a 24-km region with VW frictional properties of size $\lambda_{\text{VW}}$, surrounded by a velocity-strengthening domain. The majority of the seismic events arrest within the VW region, which we refer to as "partial ruptures," however some span the entire VW region, which we refer to as "complete ruptures" (Fig. 2C). Weakening due to thermal pressurization is confined to the region with the VW properties. The parameter values used for the simulations presented in this work are motivated by prior studies (Rice, 2006; Noda and Lapusta, 2010; Perry et al., 2020) and provided in Table 1.

## 3  Energy partitioning and notion of breakdown energy $G$

In the earthquake energy budget, the total strain energy change per unit source area $\Delta W/A$ is partitioned into the dissipated energy per unit area, $E_{\text{Diss}}/A$, and the radiated energy per unit area, $E_R/A$:

$$\Delta W/A = E_{\text{Diss}}/A + E_R/A. \tag{6}$$

The total strain energy released per unit area $\Delta W/A$ is given by:

$$\Delta W/A = \frac{1}{2}(\bar{\tau}_{\text{ini}} + \bar{\tau}_{\text{fin}})\bar{\delta}, \tag{7}$$

where $\bar{\delta}$ is the average final slip for the event, and $\bar{\tau}_{\text{ini}}$ and $\bar{\tau}_{\text{fin}}$ are the average initial and final shear stress weighted by the final slip (Noda and Lapusta, 2012), respectively,

$$\bar{\tau}_{\text{ini}} = \frac{\int_\Omega \tau_{\text{ini}}(z)\delta_{\text{fin}}(z)dz}{\int_\Omega \delta_{\text{fin}}(z)dz}, \tag{8}$$

$$\bar{\tau}_{\text{fin}} = \frac{\int_\Omega \tau_{\text{fin}}(z)\delta_{\text{fin}}(z)dz}{\int_\Omega \delta_{\text{fin}}(z)dz}. \tag{9}$$

Here, $\Omega$ represents the ruptured domain. The static stress drop is a measure of the difference in average stress before and after the rupture. The relevant definition of average static stress drop for energy considerations is the energy-based or slip-weighted stress drop (Noda et al., 2013):

$$\overline{\Delta\tau} = \bar{\tau}_{\text{ini}} - \bar{\tau}_{\text{fin}} = \frac{\int_\Omega \left[\tau_{\text{ini}}(z) - \tau_{\text{fin}}(z)\right]\delta_{\text{fin}}(z)dz}{\int_\Omega \delta_{\text{fin}}(z)dz}. \tag{10}$$

The dissipated energy per unit rupture area can be computed from the evolution of shear resistance with slip:

$$E_{\text{Diss}}/A = \frac{\int_\Omega \left[\int_0^{\delta_{\text{fin}}(z)} \tau(\delta')d\delta'\right]dz}{\int_\Omega dz}. \tag{11}$$

The dissipated energy $E_{\text{Diss}}/A$ is often further partitioned into the average breakdown energy $G$ (Palmer and Rice, 1973; Rice, 1980; Tinti et al., 2005) and residual dissipated energy (dark grey triangle and light grey rectangle in Fig. 1, respectively). The average breakdown energy represents the spatial average of the local breakdown energy $G_{\text{loc}}$ within the source region,

$$G = \frac{\int_\Omega G_{\text{loc}}(z)dz}{\int_\Omega dz} \tag{12}$$

where the local breakdown energy is defined as,

$$G_{\text{loc}}(z) = \int_0^{D_c(z)} [\tau(\delta') - \tau_{\min}(z)]d\delta', \tag{13}$$

and $\tau_{\min}(z)$ is the minimum local shear resistance during seismic slip after the initial strengthening from the initial to peak shear resistance via the direct effect. $D_c$ is defined as the critical slip distance during the rupture such that $\tau(D_c(z)) = \tau_{\min}(z)$.

Seismological studies have attempted to estimate the average breakdown energy for natural earthquakes based on the standard energy partitioning diagram (Fig. 1) as follows (Abercrombie and Rice, 2005; Rice, 2006):

$$G' = \frac{\bar{\delta}}{2}\left(\overline{\Delta\tau} - \frac{2\mu E_R}{M_0}\right), \tag{14}$$

where $G'$ is the approximation for the average breakdown energy $G$, $\overline{\delta}$ is the average slip during the rupture, $\overline{\Delta\tau}$ is the seismologically-inferred average static stress drop, $\mu$ is the shear modulus, $E_R$ is the radiated energy and $M_0$ is the seismic moment. The definition of $G'$ assumes that the rupture area exhibits negligible stress overshoot/undershoot, or that the average level of dynamic resistance during sliding is the same as the final average shear stress. Numerical studies have shown that $G'$ may indeed provide a reasonable estimate of the average breakdown energy (within a factor of 2) for crack-like ruptures, which exhibit mild overshoot/undershoot compared to the static stress drop (Perry et al., 2020), however such estimates can dramatically differ from the true values for ruptures that experience a considerable stress undershoot, as is the case of self-healing pulse-like ruptures (Lambert et al., in review).

Note that the energy balance shown in Eq. 6 reflects the energy partitioning over the rupture process as a whole. While the dissipated energy is a local quantity along the fault, the radiated energy is not and can only be related to the stress-slip behavior in the averaged sense over the entire rupture process (Fig. 1). Seismological estimates of the average breakdown energy can be made assuming the standard energy partitioning following the slip-weakening diagram (Fig. 1) and using Eq. 14 with the total radiated energy, with the results dependent on the accuracy of the radiated energy estimates and validity of the assumed energy partitioning model, which has been shown to breakdown for pulse-like ruptures (Lambert et al., in review). Estimating the local breakdown energy is more challenging. One approach is to use finite-fault slip inversions to determine the stress evolution during rupture and hence the breakdown work (e.g. Tinti et al., 2005), with the results dependent on the accuracy of finite-fault inversions that are known to be non-unique and affected by smoothing.

## 4   Breakdown energy in models with thermal pressurization of pore fluids

The local slip and stress evolution are determined at every point along the fault within our simulations at all times, thus we can calculate the local dissipation and breakdown energy throughout each rupture as well as study these quantities evolve in different ruptures throughout the sequence. We can also compute the average energy quantities and construct the average stress vs. slip curves for the total rupture process in a manner that preserves the overall energy partitioning (Noda and Lapusta, 2012). We define seismic slip to occur when the local slip velocity exceeds a velocity threshold $V_{\text{thresh}} = 0.01$ m/s. As slip rates during sliding are typically around 1 m/s or higher and drop off rapidly during the arrest of slip, modest changes of this velocity threshold by an order of magnitude produce very mild differences in $D_c$ and $G$, by less than 1%.

The average breakdown energy $G$ computed from our simulations increase with average slip and matches estimates of breakdown energy for natural events (Fig. 4), as expected from the simplified theoretical considerations (Rice, 2006). As demonstrated in previous numerical studies (Perry et al., 2020), when our fault models combine moderately efficient thermal pressurization with persistently weak conditions, such as from relatively low interseismic effective normal stresses (25 MPa) due to substantial chronic fluid overpressurization, the models produce mostly crack-like ruptures that reproduce all main observations about earthquakes, including magnitude-invariant average static stress drops of 1-10 MPa, breakdown energy values

that are quantitatively comparable to estimates from natural earthquakes, and fault temperatures well below representative equilibrium melting temperatures near $1000^o$ C for wet granitic compositions in the shallow crust (Rice, 2006). It is important to note that the presence of enhanced dynamic weakening is critical for producing reasonable values of static stress drop ($> 1$ MPa) in such fault models with chronic fluid overpressurization; otherwise, the stress changes due to the standard rate-and-state friction would be too low (as they are proportional to the effective normal stress). As such, dynamic weakening due to thermal pressurization still dominates the overall weakening behavior during dynamic rupture. These results suggest that fault models incorporating chronic fault weakness and enhanced weakening may be plausible representations of rupture behavior on mature faults. The work of Perry et al. (2020) and Lambert et al. (in review) provide a broader exploration of models with different parameters, including different levels of interseismic effective stresses and efficiency of enhanced dynamic weakening. Here, we use a representative model to illustrate the resulting properties of the breakdown energy in such models.

Let us examine the spatial distribution of shear stress and breakdown energy in three ruptures of varying size within the same simulated sequence of earthquakes (Fig. 5). All three ruptures nucleate, propagate, and arrest predominantly in the VW region that has uniform fault properties, the only difference being how big the events become. The distribution of shear stress along the fault before each rupture is heterogeneous due to the stress drop from previous ruptures. While each earthquake nucleates in a region with approximately the same locally-high initial stress, the ruptures propagate and arrest over regions with lower prestress. Larger ruptures with more slip experience greater weakening and larger local stress drops in some regions, which facilitates further rupture propagation over areas of lower prestress. As such, while the final average shear stress decreases for larger ruptures, the average initial stress also decreases, resulting in nearly magnitude-invariant average stress drops.

Despite the fault constitutive properties being uniform and constant in time, the breakdown energy varies spatially within each event as well as differs at each location for different ruptures (Figs. 5C and 6). Larger ruptures that experience larger average slip also exhibit more weakening, resulting in the average breakdown energy generally increasing with the rupture size (Fig. 5C). If we examine individual points that are common among all three ruptures, we see that the local breakdown energy also varies as the points experiences different degrees of slip and overall weakening behavior (Fig. 6). This suggests that the local and average breakdown energy is not just a function of the local fault material properties, but a more complicated evolution of effective weakening behavior and stress throughout the rupture.

Note that the breakdown energy illustrated in Fig. 6 is dominated by the thermal pressurization of pore fluids, with negligible contribution from the weakening due to standard rate-and-state friction. The breakdown energy due to rate-and-state friction can be estimated as (Perry et al., 2020):

$$G = \frac{1}{2} b \overline{\sigma} D_{\text{RS}} \left( \log \frac{\theta_{\text{ini}} V_{\text{dyn}}}{D_{\text{RS}}} \right)^2 \tag{15}$$

where the effective normal stress $\overline{\sigma}$ is assumed to be constant, $\theta_{\text{ini}}$ is the value of the state variable at the beginning of slip, and $V_{\text{dyn}}$ is the representative dynamic slip rate. Assuming that $\overline{\sigma}$ is still approximately given by the interseismic value at the

beginning of slip (which would produce a upper bound), $\theta_{\text{ini}}$ is given by the representative inter-event time of 10 years, and $V_{\text{dyn}}$ is given by the representative peak rate of 10 m/s, the breakdown energy due to the standard rate-and-state friction in our simulation has the upper bound of 0.15 MJ/m$^2$. This is an order of magnitude smaller than the values of 1 to 6 MJ/m$^2$ of Fig. 6.

## 5   Overall increase of breakdown energy with slip and significant rupture-dependent scatter

Previous theoretical work has demonstrated how the incorporation of thermo-hydro-mechanical processes such as the thermal pressurization of pore fluids can explain the inferred increase in breakdown energy with increasing event size (Rice, 2006). The work of Rice (2006) presented solutions for two end-member cases for the evolution of shear resistance and breakdown energy with thermal pressurization, illustrating how continuous weakening occurs with slip and results in breakdown energy increasing with slip.

If slip occurs within a layer of thickness $h$ that is large enough to justify the neglect of heat and fluid transport, conditions may be considered adiabatic and undrained, which may be relevant for relatively short slip durations (Rice, 2006; Viesca and Garagash, 2015). Under such conditions, the weakening behavior is controlled by the ratio of the coupling coefficient $\Lambda$ and specific heat $\rho c$, as well as the thickness of the shearing layer $h$ which controls the efficiency of heat production. Assuming a constant friction coefficient $f$ and slip rate $V$, one can express the evolution of shear resistance $\tau$ and breakdown energy $G$ as functions of slip (Rice, 2006),

$$\tau(\delta) = f\left(\sigma - p_0\right)\exp\left(-\frac{f\Lambda}{\rho c}\frac{\delta}{h}\right), \tag{16}$$

$$G(\delta) = \frac{\rho c\left(\sigma - p_0\right)h}{\Lambda}\left[1 - \left(1 + \frac{f\Lambda\delta}{\rho ch}\right)\exp\left(-\frac{f\Lambda\delta}{\rho ch}\right)\right]. \tag{17}$$

Under such conditions, increasing slip results in continued weakening of the shear resistance and increasing values of breakdown energy. The continued weakening is the result of shear heating and subsequent pressurization, which remains active as long as the slip rate and shear stress are non-zero.

The inclusion of thermal and hydraulic diffusion introduces a diffusion time-scale to the problem, which governs the efficiency of weakening over extended slip durations. If one considers slip on a mathematical plane, a characteristic weakening time-scale $t^*$, may be defined assuming a constant friction coefficient and slip rate (Mase and Smith, 1987):

$$t^* = \frac{4}{f^2}\left(\frac{\rho c}{\Lambda}\right)^2\frac{\left(\sqrt{\alpha_{\text{hy}}} + \sqrt{\alpha_{\text{th}}}\right)^2}{V^2}. \tag{18}$$

Rice (2006) demonstrated that this may be related to a characteristic slip-weakening distance for thermal pressurization,

$$L^* = \frac{4}{f^2}\left(\frac{\rho c}{\Lambda}\right)^2\frac{\left(\sqrt{\alpha_{\text{hy}}} + \sqrt{\alpha_{\text{th}}}\right)^2}{V}, \tag{19}$$

such that the evolution of shear resistance and breakdown energy for slip on a plane may also be expressed as a function of slip (Rice, 2006):

$$\tau(\delta) = f(\sigma - p_0)\exp\left(\frac{\delta}{L^*}\right)\operatorname{erfc}\left(\sqrt{\frac{\delta}{L^*}}\right), \tag{20}$$

$$G(\delta) = f(\sigma - p_0)L^*\left[\exp\left(\frac{\delta}{L^*}\right)\operatorname{erfc}\left(\sqrt{\frac{\delta}{L^*}}\right)\left(1 - \frac{\delta}{L^*}\right) - 1 + 2\sqrt{\frac{\delta}{\pi L^*}}\right]. \tag{21}$$

Unlike the case of a critical slip-weakening distance $D_c$ in standard slip-weakening models, the weakening of shear resistance is continuous with increasing slip (Fig. 7a), with $L^*$ providing a measure of how much slip is needed to weaken by a certain degree. Note that the evolution of stress in Eqs. (16) and (20) do not consider the elastic interactions that occur due to non-uniform slip within finite ruptures, and therefore assume that the slip velocity is not only temporally constant, but spatially uniform over the fault.

Both of these thermal pressurization solutions have the convenient feature of expressing the breakdown of shear resistance as a function of slip, drawing familiarity to standard slip-weakening notions of shear fracture. As pointed out by Rice (2006), the representation of breakdown energy purely as a function of slip is a considerable simplification, whereas the physics underlying the mechanisms for weakening require that $\tau$ is a complicated function of the slip rate history up to the current time. During dynamic rupture, the local slip rate experiences considerable acceleration near the rupture front, resulting in a more pronounced weakening rate (Fig. 7), which in turn facilitates large dynamic stresses and higher slip rates in other parts of the rupture. As the rupture front passes, both the slip rate and weakening rate decrease. However, the slip rate may persists around typical seismic values of 1 m/s until the arrival of arrest waves from the edges of the rupture or local healing. Note that while the slip rates behind the rupture front in our models appear more or less stable around 1 m/s (Fig. 7E and G), they may vary depending on the arrival of wave-mediated dynamic stresses from other slipping regions in the rupture, which drive prolonged slip and therefore modulate the weakening rate due to shear heating mechanisms like thermal pressurization. In general, the friction coefficient may also vary considerably with the slip rate, particularly when accounting for additional enhanced weakening processes such as flash heating (Rice, 1999; Goldsby and Tullis, 2011; Passelegue et al., 2014).

The continued weakening with slip due to thermal pressurization is an important factor that drives rupture propagation and allows ruptures to propagate under lower, and hence less favorable, prestress conditions. Let us consider two fault models with the same initial prestress and the same rate-and-state frictional parameters, but with and without enhanced weakening due to thermal pressurization (Fig. 8). The rupture governed by only standard rate-and-state friction exhibits relatively mild stress variations with slip rate and thus requires higher prestress conditions to propagate. While the local slip rate evolution varies among points throughout the rupture, the evolution of shear resistance with slip associated with the breakdown process is generally comparable throughout the rupture with uniform rate-and-state properties (Fig. 8 left column). In contrast, the rupture that is driven by enhanced weakening due to thermal pressurization experiences a stronger feedback between the evolution of shear stress and slip rate, resulting in a much larger rupture that propagates over lower prestress conditions. The evolution of

slip rate is highly variable for different points throughout the crack-like rupture, with long tails of seismic slip behind the rupture front that experience periods of acceleration and deceleration due to dynamic stress interactions from neighboring points.

This variability in local slip rate translates into further variability in local weakening, even for points with the same initial prestress. This emphasizes that the local weakening behavior, and the associated breakdown energy, depend not only on the local prestress and weakening properties, but also the distribution of prestress and weakening behavior throughout the entire rupture process.

An important consequence of continued fault weakening is that much of the additional dissipated energy, which leads to the increase of breakdown energy with continued slip, is not concentrated near the rupture front (Fig. 7). Moreover, weakening may not actually be strictly monotonic but local points can experience transient increases in shear stress as they begin to arrest, but then are loaded by neighboring slipping regions and forced to slip and weaken further (Fig. 6 and 10). The continued and variable weakening of shear resistance behind the rupture front emphasizes a critical difference between dynamic shear ruptures and mode I fracture, where the crack surface is typically traction-free behind the cohesive zone at the rupture front. The attribution of the continually dissipated energy to the breakdown process governing rupture propagation is also inconsistent with the assumption of small-scale yielding, which facilitated the original mathematical analogy based on laboratory constitutive relations derived at lower slip-rates (Palmer and Rice, 1973).

While breakdown energy does not appear to be a constant material property, one may ask if the effects of local weakening due to thermal pressurization may be adequately encapsulated into a slip-weakening formulation such as Eqs. (16-20). To gain insight into such possibility, let us examine three large ruptures in our simulations that have comparable average slip and breakdown energy (Fig. 9). If we consider the evolution of local shear stress and slip at points shared among the three ruptures, we can see that the local breakdown energy differs even for comparable local slip. Moreover, the three points, which share the

same constitutive description, do not exhibit a systematic scaling relationship between local slip and breakdown energy. For example, the point at $z = -4.8$ km exhibits a generally increasing trend in local $G$ with increasing slip, whereas the point at $z = 4.8$ km shows decreasing values of $G$ for increasing slip among the three ruptures (Fig. 9C vs. E). The point in the center of the rupture ($z = 0$) does not even exhibit a monotonic trend, as G both increases and decreases for ruptures with increasing slip (Fig. 9D). Indeed, if we examine the spatial distribution of local stress and breakdown energy within each rupture, we see

that while the three ruptures have comparable average $G$ and slip, they achieve both in different ways (Fig. 10).

The general trend of increasing breakdown energy with slip qualitatively holds for most local points within our simulated ruptures, however there is considerable variability for individual values of $G$ at a given slip (Fig. 11). While values of average breakdown energy and slip for individual ruptures appear to demonstrate a consistent scaling relationship, these average values

smooth out the greater variability in local breakdown energy and slip. For points within our simulated ruptures that experience a net decrease, or breakdown, in shear stress, the local G is generally within a factor of 3 of the scaling relationship between average $G$ and average slip. This variation adds up to approximately an order of magnitude variation in local $G$ for some values

of slip.

For frictional ruptures, substantial slip may occur in regions that experience a net increase in shear stress, particularly in the regions near the rupture arrest (Fig. 6B). We find that points in our simulated ruptures that experience a net increase in shear stress exhibit greater variability in $G$ with slip (Fig. 11, yellow circles), potentially due to the greater variability in slip rate during rupture deceleration and arrest. These points illustrate the challenge of partitioning the dissipated energy into components that are thought to be, and not be, relevant to the dynamic rupture process. These points exhibit no net local breakdown of shear resistance but rather a net strengthening. A more appropriate approach may be to distinguish between concepts of breakdown energy and "restrengthening energy," as discussed in Tinti et al. (2005). However, the physical relevance for either component, or their distinction, during the rupture process is not directly evident. Understanding the physical significance of different components of dissipated energy for dynamic rupture propagation is an important topic of active research.

The theoretical considerations of Rice (2006) have been extended to the spatially and temporally variable slip rate associated with steady rupture propagation (Viesca and Garagash, 2015). Approximate expressions for the scaling of breakdown energy with slip can be presented for end-member conditions of undrained $G_{\mathrm{u}}(\delta)$ and drained $G_{\mathrm{d}}(\delta)$ weakening as:

$$G_{\mathrm{u}}(\delta) \approx f(\sigma - p_0)\frac{f\Lambda\delta^2}{2\rho ch}, \qquad\qquad \text{undrained, small slip} \qquad\qquad (22)$$

$$G_{\mathrm{d}}(\delta) \approx (12\pi)^{-1/3}f(\sigma - p_0)L^{*1/3}\delta^{2/3}, \qquad \text{slip on a plane, large slip.} \qquad\qquad (23)$$

Similar to the solutions (17) and (21) that assume constant slip rate, the steady-state solutions (22-23) do not capture the variability of the local breakdown energy with slip seen in our simulated dynamic ruptures (Fig. 11). This is because our simulated dynamic ruptures do not exhibit steady rupture propagation, but rather have considerable spatial variations in slip rate evolution, as likely the case for natural earthquake ruptures. This comparison illustrates a limitation of steady-state rupture solutions for examining rupture properties that are highly sensitive to spatial heterogeneity in slip motion, such as breakdown energy in the presence of thermal pressurization.

While the general increase in breakdown energy with slip is qualitatively consistent among the theoretical solutions and our simulated dynamic ruptures in 2D models with 1D faults (Fig. 11), the scaling relationship between breakdown energy and slip would be best studied in 3D models of dynamic rupture with 2D faults. For example, for ruptures on 2D faults would have a larger fraction of the ruptured area associated with rupture arrest and hence may demonstrate a wider scatter in local $G$, as seen by points in our simulated ruptures that experience a net increase in shear stress. In addition, it would be prudent to examine any differences in scaling behavior for ruptures that are geometrically confined along a given direction, as may be representative of large crustal earthquakes. However, we expect that the main results of this work - that the local and average breakdown energy can vary among ruptures and are not unique functions of slip - would be consistent with 2D rupture scenariios in 3D models.

## 6 Conclusions

The average breakdown energy for our simulated ruptures tends to increase with increasing rupture size and average slip in a manner consistent with inferences from field observations and simplified theoretical models (Rice, 2006; Viesca and Garagash, 2015). At the same time, the values of local breakdown energy for a given amount of slip have a wide spread in our simulations, even though the constitutive properties are uniform and time-independent along the fault, highlighting the reality that breakdown energy in models with thermo-hydro-mechanical mechanisms is not fundamentally a function of slip. In fact, ruptures with near-uniform slip can have local values of the breakdown energy vary by as much as a factor of 4 (Fig. 10C), making a homogeneous fault appear to be heterogeneous. This is because the breakdown energy depends on the specific history of motion and dynamic stress changes that occur throughout individual rupture processes. Furthermore, since the history of rupture motion is determined, in part, by the fault prestress before the dynamic rupture, the breakdown energy also depends on the history of other slip events on the fault that determine the prestress.

The analytic formulations for the evolution of shear resistance with slip for the thermal pressurization presented by Rice (2006) provide profound insight into the first-order behavior of such thermally-activated hydro-mechanical weakening mechanisms. However, they are based on the kinematic assumptions of a *spatially uniform and temporally constant* slip velocity, as well as a constant friction coefficient, that allow for the weakening rate to be determined as a function of slip. In the fully dynamic statement of the problem, the evolving and spatially non-uniform slip rate is a key part of the solution which leads to the evolution in the associated shear heating and weakening/strengthening of the fault that depend not only on the amount of slip but also on how that slip is achieved through the complex history of slip velocity. Our results demonstrate that the extension to steady-state rupture solutions with non-constant slip rate (Viesca and Garagash, 2015) similarly does not capture the variability in local breakdown energy associated with the complex and evolving history of slip velocity and dynamic stress interactions in non-steady ruptures, even in fault models with uniform fault properties like ours.

Note that this variability in local $G$ for a given slip is achieved among points with uniform and constant constitutive properties. Such variability in the effective weakening rate and $G$ may become more pronounced in the presence of fault heterogeneity, such as for geometrically rough faults with variable effective normal stress, or if the hydraulic properties of the shearing layer and surrounding rock were to evolve during the rupture process, such as from changes in rock permeability due to off-fault damage. The evolution of permeability during dynamic rupture may have considerable implications for the role of thermo-hydro-mechanical processes in the evolution of shear resistance on faults and it is an important topic for future work.

While we follow the assumption that most of the breakdown energy occurs on the shearing surface (Rice, 2006; Viesca and Garagash, 2015), additional dissipation may also come from the production of damage and off-fault inelastic deformation (Poliakov et al., 2002; Andrews, 2005; Okubo et al., 2019), especially on rough, non-planar faults (Dunham et al., 2011b). Such sources of additional dissipated energy may contribute to the inferred increase in average breakdown energy with average slip

for natural earthquakes. Estimates from laboratory and field measurements suggest that the contribution of damage and other off-fault processes to dissipation may be relatively small, <10 % (Chester et al., 2005; Rockwell et al., 2009; Aben et al., 2019), however, this remains an area of active research. Since the off-fault damage would be rupture-dependent as well, adding it to the consideration of the breakdown energy would likely further reinforce the conclusion of this study that breakdown energy is not an intrinsic fault property but rather is rupture-dependent.

The finding that the breakdown energy - as well as the weakening rate - can vary substantially along a given rupture and among subsequent ruptures, even for comparable values of slip, suggests that caution is needed in using the inferred breakdown energies from natural events for modeling of future earthquake scenarios. Some dynamic rupture simulations account for thermo-hydro-mechanical effects (Andrews, 2002; Bizzarri and Cocco, 2006; Noda et al., 2009; Schmitt et al., 2015) and/or in-
455 corporate the effects of inelastic off-fault damage (Dunham et al., 2011a, b; Roten et al., 2017; Withers et al., 2018) that should result in qualitatively similar effects on the breakdown energy. However, many employ simplified shear resistance evolutions that prescribe the breakdown energy and/or weakening rate directly, as a local fault property (Richards-Dinger and Dieterich, 2012; Shaw et al., 2018; Gallovic et al., 2019; Dalguer et al., 2020). Future work is needed to investigate whether and how the complexity of the local weakening/strengthening behavior experienced by the simulated faults with thermo-hydro-mechanical
and other mechanismscan be translated into simulations with more simplified local relations, e.g. slip-dependent, and still result in similar rupture dynamics.

Furthermore, several features of faulting in the presence of thermo-hydro-mechanical effects call into question the overall analogy with cohesive-zone dynamic fracture theory and hence the significance of the breakdown energy as the quantity that
controls rupture dynamics. The analogy between breakdown and fracture energies, and more broadly frictional faulting and shear cracks of traditional fracture mechanics, requires that the breakdown process be confined close to the rupture tip (small-scale yielding) and that the dynamic resistance level be constant; under such conditions, the conclusions of dynamic fracture theory apply, including on the significance of breakdown energy (Freund, 1990). However, neither of these assumptions holds for the faults with thermo-hydro-mechanical processes. The weakening - and hence breakdown process - typically continues
with ongoing slip at seismic slip rates on such faults, long after the rupture front passes. As a result, the breakdown process is not confined to the rupture tip and the dynamic resistance level is not constant. Moreover, the total dissipated energy - not just the energy included in the notion of breakdown energy - contributes to shear heating and hence fault weakening in thermo-hydro-mechanical fault models. That is why the entire dissipated energy may affect rupture dynamics as well. These considerations emphasize the need for better understanding of rupture dynamics and its controls in the presence of thermo-hydro-mechanical
processes and for more systematic incorporation of such processes in earthquake source modeling.

*Data availability.* The data supporting the analysis and conclusions are accessible through the CaltechDATA repository ( https://data.caltech.edu/records/1447 ).

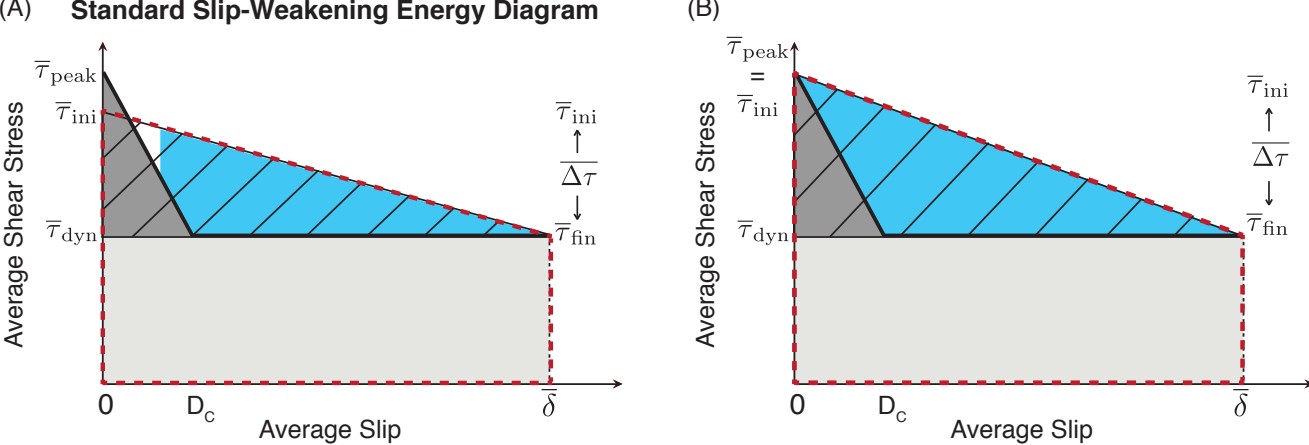

**Figure 1.** (A) Standard linear slip-weakening diagram where the average shear stress is assumed to increase from an initial to peak stress with no slip and then linearly decrease to a dynamic resistance level over a critical slip distance $D_c$. The difference between the average initial and final shear stress levels is called the static stress drop. The average stress vs. slip diagram is used to represent the energy partitioning of the total strain energy change per unit rupture area (dashed red trapezoid) into the breakdown energy (dark grey triangle), residual dissipated energy per unit area (light gray rectangle), and radiated energy per unit area (blue region). The additional dissipation associated with the initial strengthening outside of the red trapezoid comes at expense of the radiated energy (white triangle inside the red-dashed trapezoid). (B) The case of the initial stress equal to the peak stress. Note that this diagram is an approximation even if the local behavior is governed by linear slip-weakening friction, since different points of the rupture would have different slip, including near-zero slip close to the rupture edges, and averaging over the dynamic rupture would produce a different curve from the local behavior (Noda and Lapusta, 2012).

**Table 1.** Model parameters used in simulations of earthquakes and aseismic slip.

| Parameter | Symbol | Value |
|---|---|---|
| Loading slip rate | $V_{\mathrm{pl}}$ | $10^{-9}$ m/s |
| Shear wave speed | $c_s$ | 3299 m/s |
| Shear modulus | $\mu$ | 36 GPa |
| *Rate-and-state parameters* | | |
| Reference slip velocity | $V_*$ | $10^{-6}$ m/s |
| Reference friction coefficient | $f_*$ | 0.6 |
| Characteristic slip | $D_{\mathrm{RS}}$ | 1 mm |
| Rate-and-state direct effect (VW) | $a$ | 0.010 |
| Rate-and-state evolution effect (VW) | $b$ | 0.015 |
| Rate-and-state direct effect (VS) | $a$ | 0.050 |
| Rate-and-state evolution effect (VS) | $b$ | 0.003 |
| *Thermal pressurization parameters* | | |
| Interseismic effective normal stress | $\bar{\sigma} = (\sigma - p)$ | 25 MPa |
| Coupling coefficient (when TP present) | $\Lambda$ | 0.34 MPa/K |
| Thermal diffusivity | $\alpha_{\mathrm{th}}$ | $10^{-6}$ m$^2$/s |
| Hydraulic diffusivity | $\alpha_{\mathrm{hy}}$ | $10^{-3}$ m$^2$/s |
| Specific heat | $\rho c$ | 2.7 MPa/K |
| Shear zone half-width | $w$ | 10 mm |
| *Length scales* | | |
| Fault length | $\lambda$ | 96 km |
| Frictional domain | $\lambda_{\mathrm{fr}}$ | 72 km |
| Velocity-weakening region | $\lambda_{\mathrm{VW}}$ | 24 km |
| Cell size | $\Delta z$ | 3.3 m |
| Quasi-static cohesive zone | $\Lambda_0$ | 75 m |
| Nucleation size (Rice & Ruina, 1983) | $h_{RR}^*$ | 200 m |
| Nucleation size (Rubin & Ampuero, 2005) | $h_{RA}^*$ | 490 m |

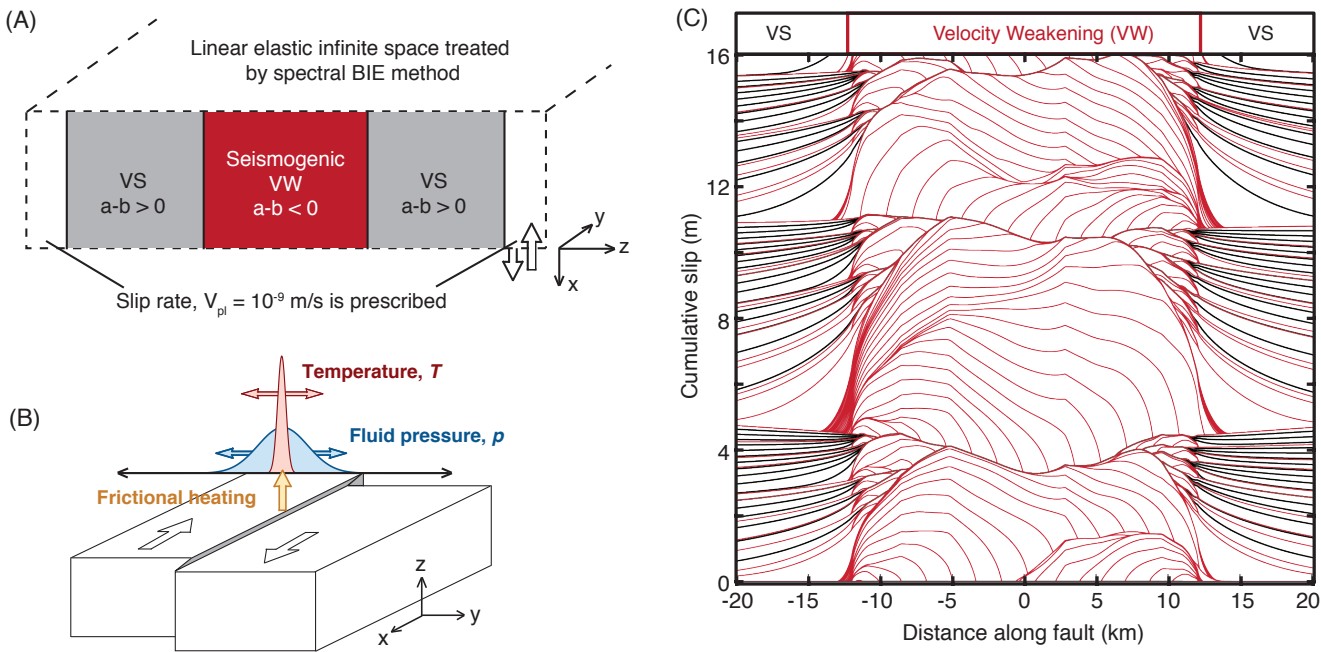

**Figure 2.** (A) The fault model incorporates a velocity-weakening (VW) seismogenic region surrounded by two velocity-strengthening (VS) sections. A fixed plate rate is prescribed outside of these regions. (B) We incorporate enhanced dynamic weakening due to the thermal pressurization of pore fluids by calculating the evolution of temperature and pore fluid pressure due to shear heating and off-fault diffusion throughout our simulations. (C) The beginning of the accumulated slip history for simulated sequences of crack-like earthquake ruptures and aseismic slip. Seismic events are illustrated by red lines with slip contours being plotted every 0.5 seconds while interseismic slip is plotted in black every 10 years. The total simulated slip history spans 2675 years corresponding to cumulative slip of 84 m and contains 200 seismic events.

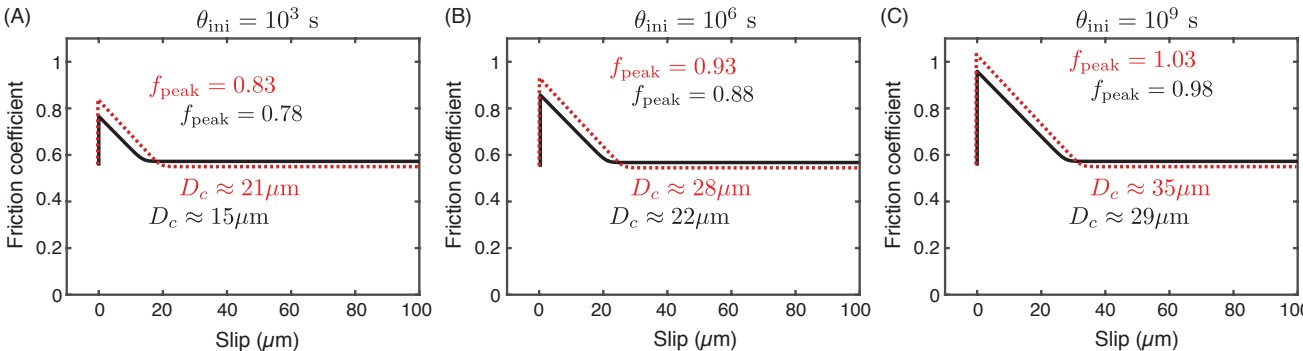

**Figure 3.** Illustration of the rate- and state-dependence of peak and dynamic friction coefficients, $f_{\text{peak}}$ and $f_{\text{dyn}}$ respectively, as well as the effective slip-weakening distance $D_c$. (A-C) Evolution of friction coefficient with slip for points with the same initial friction coefficient of 0.58 but different values of initial state variable $\theta_{\text{ini}}$, corresponding to different histories of previous motion. The initially locked point slips at an imposed slip rate of V = 1 cm/s (black) or V = 1 m/s (red), to approximately reproduce transition from the locked state to dynamic sliding as the rupture propagates through. For a given slip rate, the friction evolves to a new steady-state level, $f_{\text{dyn}} = 0.54$ and $f_{\text{dyn}} = 0.56$ for V = 1 m/s and V = 1 cm/s, respectively. These levels are similar, as expected from the logarithmic dependence on the slip rate and a narrow range of dynamic slip rates. The peak friction coefficient and effective slip-weakening distance vary more significantly with $\theta_{\text{ini}}$, where the peak friction coefficient increases for higher $\theta_{\text{ini}}$ associated with longer inter-event healing times. The example uses typical laboratory values of $(a - b) = 0.004$, $f_* = 0.6$, $L = 1\mu m$, and $V_* = 10^{-6}$ m/s.

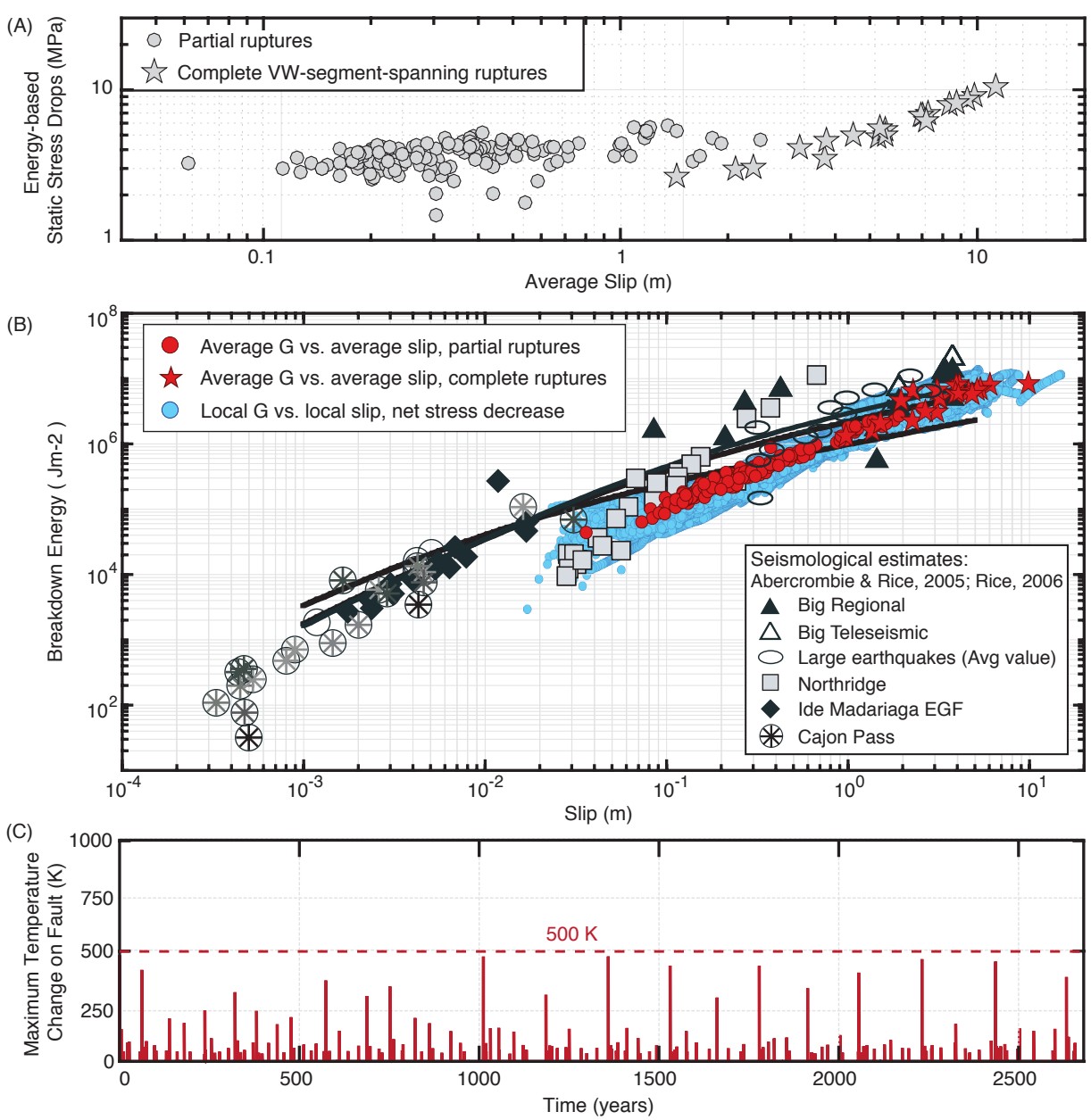

**Figure 4.** (A) The simulations result in a sequence of mostly crack-like ruptures that, despite including dynamic weakening due to thermal pressurization of pore fluids, are capable of reproducing nearly-magnitude invariant average static stress drops, with values between 1 - 10 MPa. (B) These crack-like ruptures display the overall increasing trend in average breakdown energy with average slip, as inferred for natural earthquakes (Abercrombie and Rice, 2005; Rice, 2006). (C) The simulated fault maintains reasonable temperatures and avoids melting, due to relatively low interseismic effective normal stress of 25 MPa (and hence chronic fluid overpressurization) and sufficiently efficient enhanced weakening due to thermal pressurization of pore fluids.

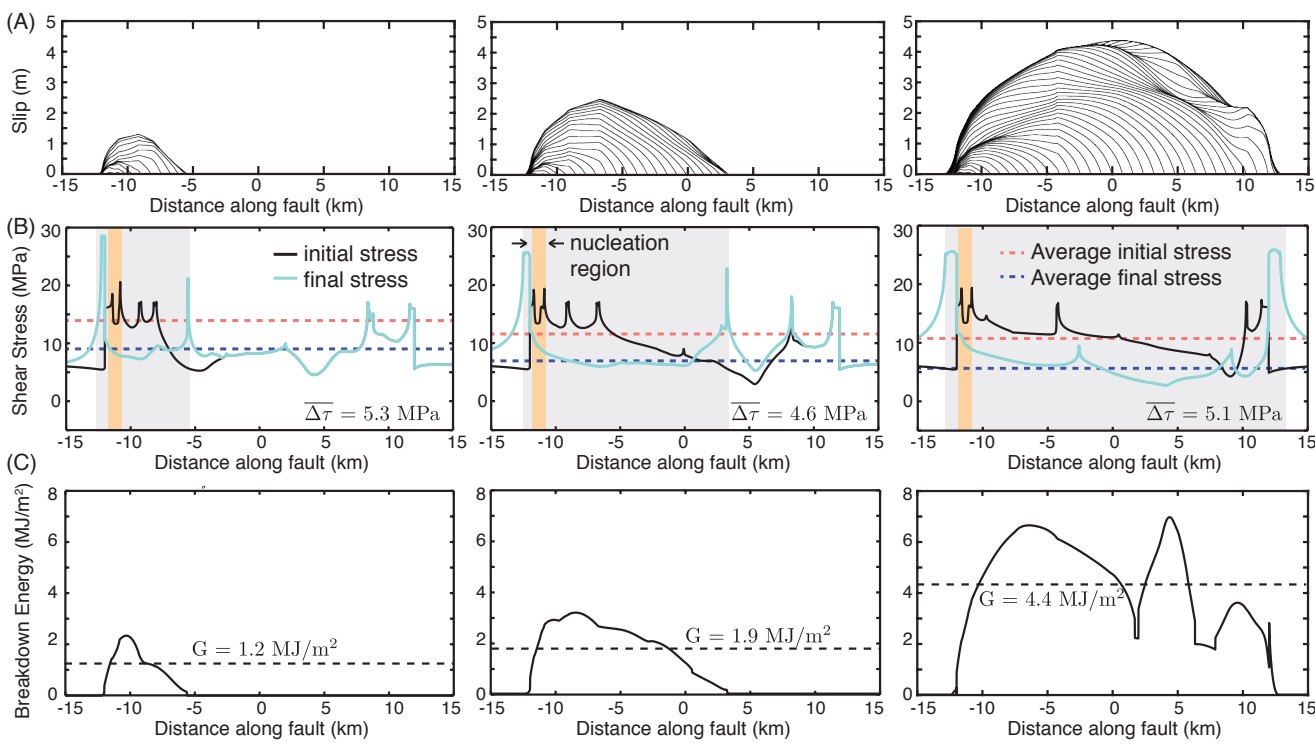

**Figure 5.** Comparison of three earthquake ruptures of different sizes nucleating over the same fault area. (A) Slip distributions for the three ruptures. (B) Distributions of initial (solid black) and final (solid blue) shear stress for the three ruptures. Gray shading denotes the ruptured region and orange shading denotes the region where each rupture nucleates. The dashed red and blue lines denote the average initial and final shear stress in the ruptured region. Large events have smaller initial and smaller final average stress, resulting in similar stress drops. (C) Distribution of breakdown energy (solid black) and average breakdown energy for each event (dashed line). The average breakdown energy generally increases with the rupture size.

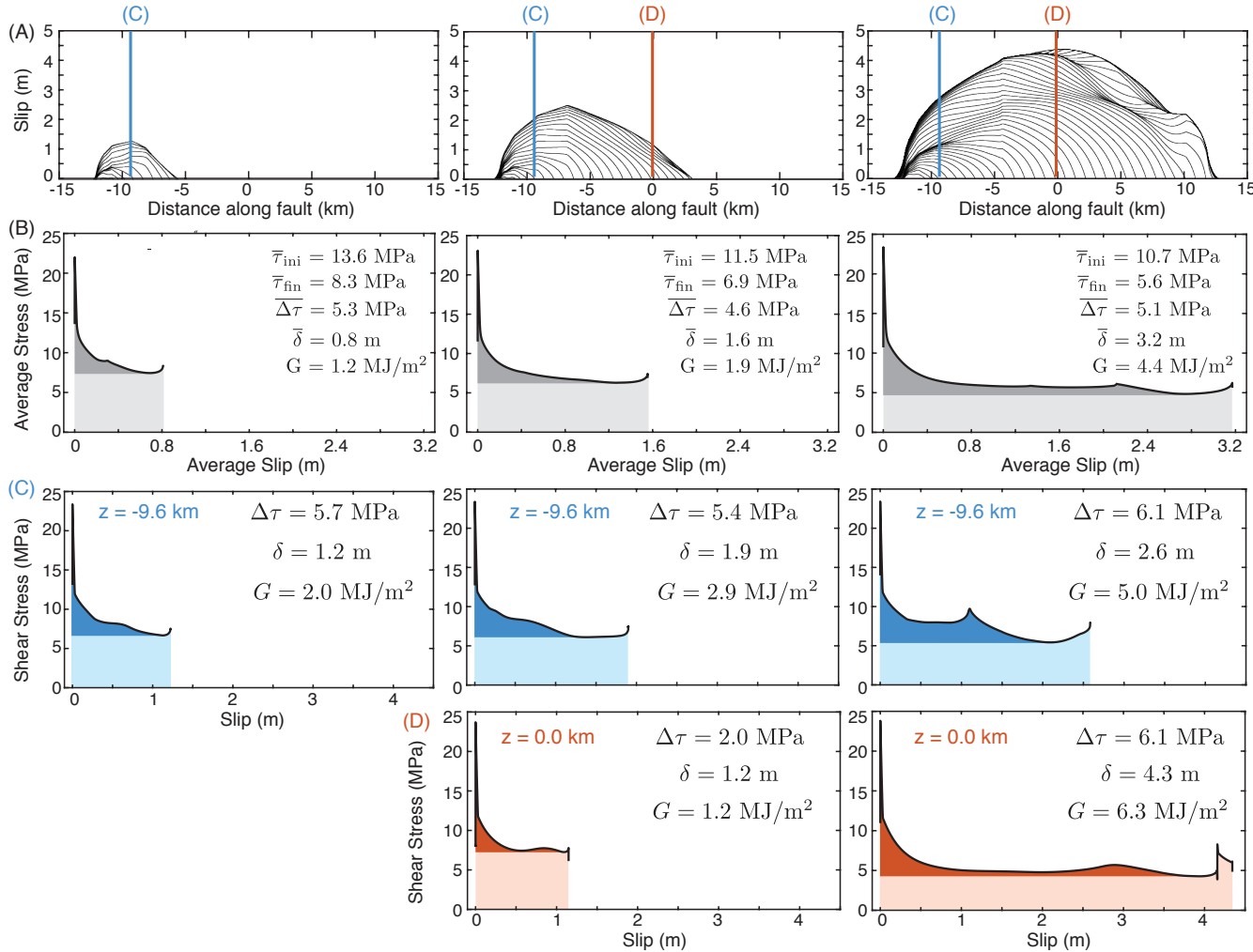

**Figure 6.** The dependence of shear stress on slip for the three ruptures of Figure 5. A) Slip distributions with locations examined in detail marked. (B) Average shear stress versus slip curves illustrating the energy partitioning of the ruptures, based on the averaging methodology of Noda and Lapusta (2012) that attempts to preserve local rupture behavior. The curves capture the continuous weakening with slip experienced by most rupture locations. (C-D) Local shear stress versus slip curves at two points within the three ruptures, illustrating the general trend in increasing breakdown energy with increasing slip at the same point.

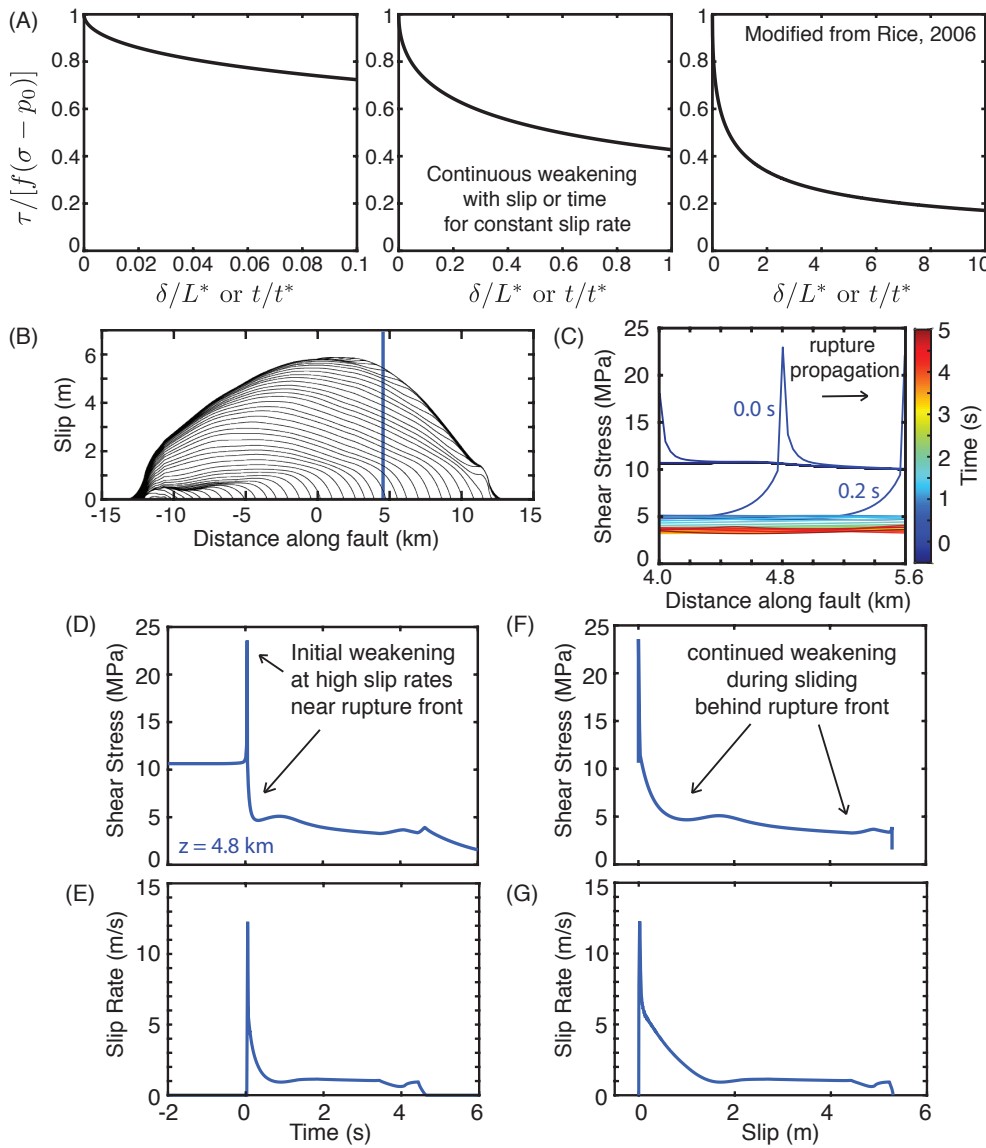

**Figure 7.** (A) Prediction of continuous weakening of shear resistance with slip or time due to the thermal pressurization of pore fluids during slip on a plane at constant slip rate V and constant friction coefficient $f$ (Rice, 2006). (B) Evolution of slip during a dynamic rupture, slip contoured every 0.2 s. (C) Evolution of shear stress localized around the point $z = 4.8$ km within the rupture. The time window shown corresponds to the duration of sliding at seismic slip rates at $z = 4.8$ m. (D-E) Evolution of local shear stress and slip rate with time at the point indicated by the blue line in (B). (F-G) Evolution of local shear stress and slip rate with slip at the same point. While qualitatively consistent with (A) in terms of the continued weakening with slip and time, the evolution of shear resistance during dynamic ruptures depends on the more complicated history of slip rate, which varies throughout the rupture process. Most of the initial local weakening occurs at slip rates higher than 1 m/s as the rupture front passes by, followed by more gradual weakening behind the rupture front at lower, but still seismic, slip rates.

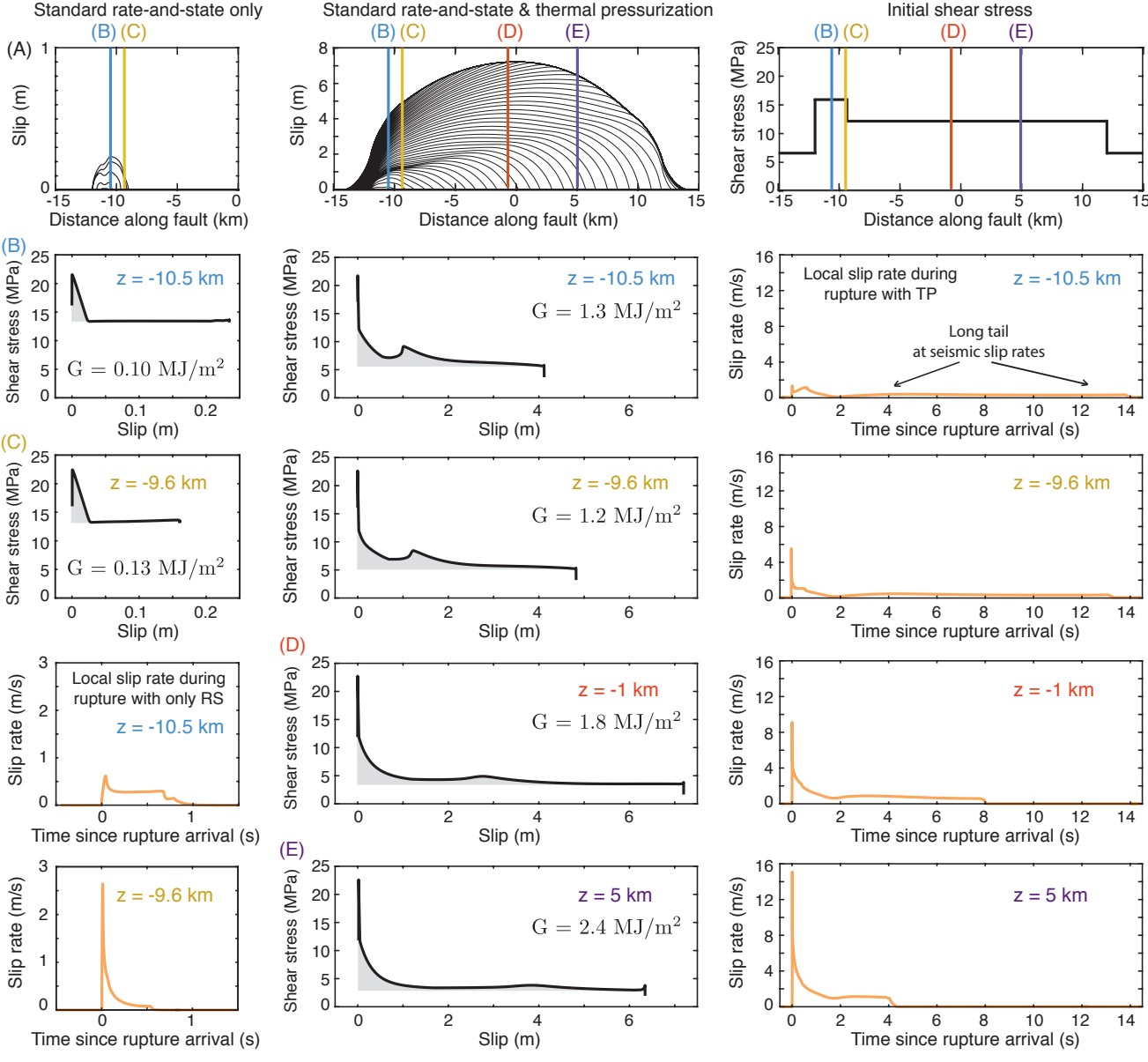

**Figure 8.** Comparison of accumulated slip, local shear stress vs. slip and local slip rate vs. time for ruptures with rate-and-state (RS) friction with and without enhanced weakening due to thermal pressurization (TP). The two ruptures occur with the same initial shear stress distribution (top right), which results in a relatively small rupture in the RS-only model that is localized within the relatively highly prestressed nucleation region (top left) The inclusion of TP allows the rupture to grow and propagate over lower prestress conditions (top center). (Left column) For the rupture governed by only RS, the breakdown of shear resistance is generally comparable at different locations with the same fault properties, despite differences in local slip rate. This is due to the relatively mild dependence of RS friction on slip rate. (Center and right columns) The rupture governed by RS and TP exhibits a more complex evolution of local shear stress and slip rate throughout the rupture, which depends not only on the local prestress but also the prestress and weakening behavior over the entire rupture through dynamic stress interactions.

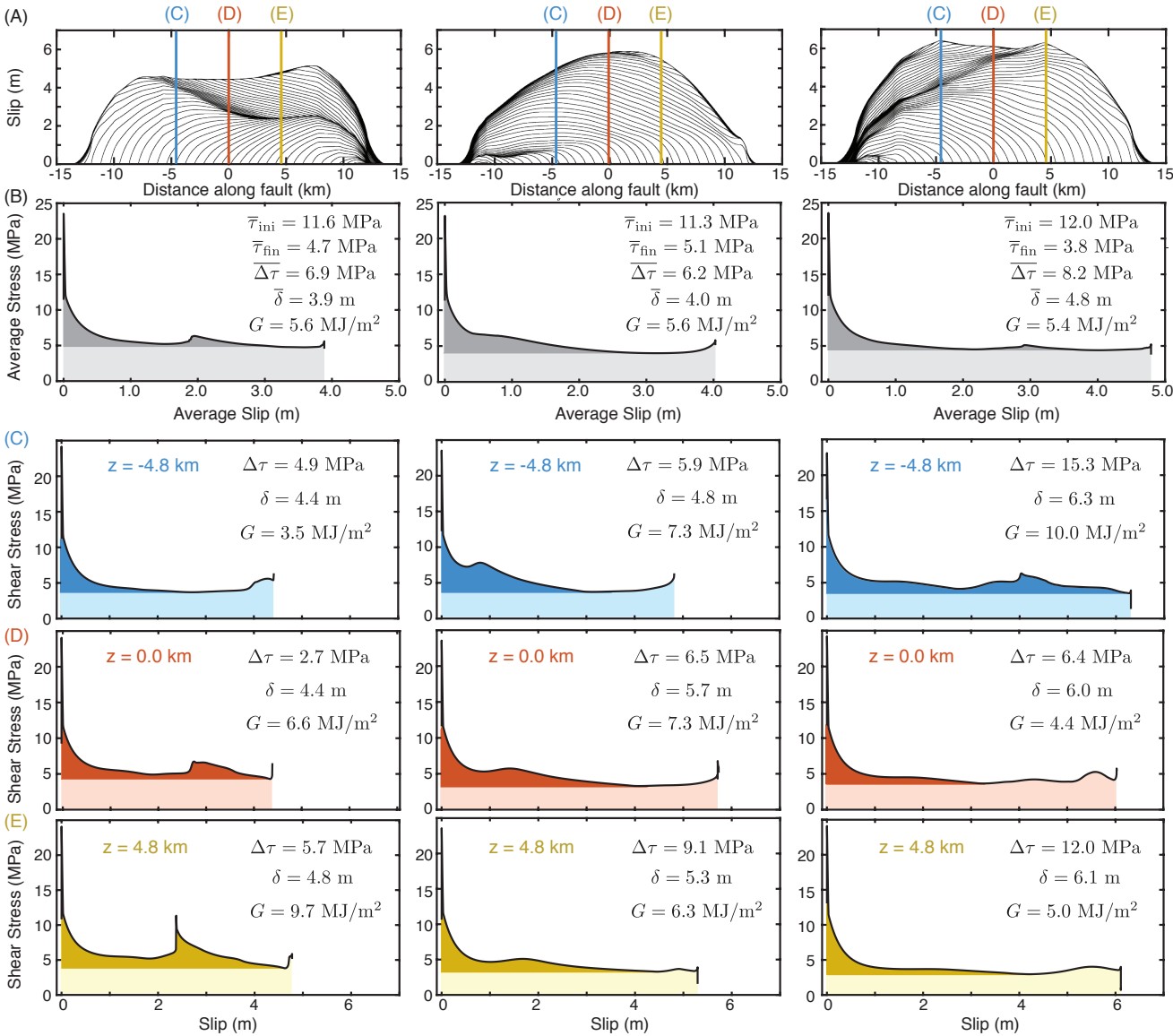

**Figure 9.** Comparison of local breakdown energy for three large earthquake ruptures with nearly the same average breakdown energy and comparable average slip. (A) Slip distributions for the three ruptures. (B) Average shear stress versus slip curves illustrating the energy partitioning of the ruptures. (C-E) Local shear stress versus slip curves at three points within the ruptures. There is not a strictly increasing trend of breakdown energy with slip for all points. In (C), the point $z = -4.8$ km experiences increasing $G$ with increasing slip. However, in (E), the point $z = 4.8$ km experiences lower values of $G$ in ruptures with larger local slip.

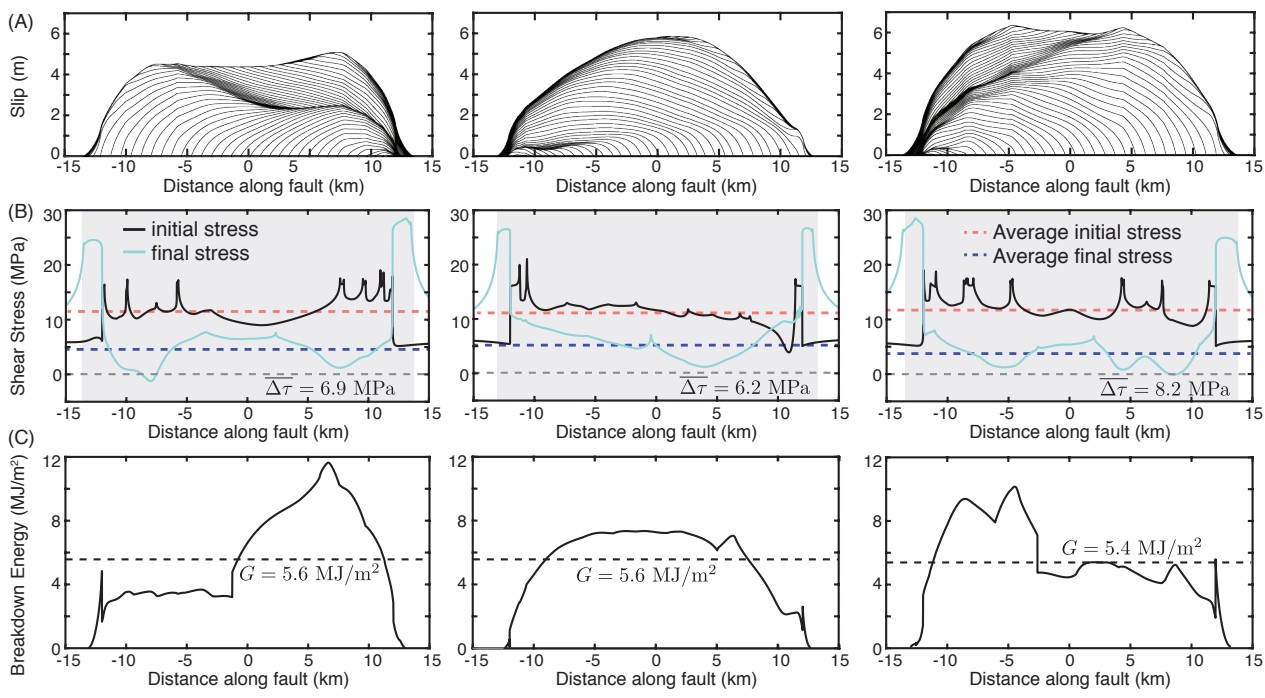

**Figure 10.** Comparison of the spatial breakdown energy distribution for the three large earthquake ruptures with nearly the same average breakdown energy and comparable average slip of Fig. 9. (A) Slip distributions for the three ruptures. (B) Spatial distributions of initial (solid black) and final (solid blue) shear stress for the three ruptures. Gray shading denotes the ruptured region and dashed red and blue lines indicate the average initial and final shear stresses, respectively. (C) Spatial distributions of the local breakdown energy. While the three ruptures have comparable average breakdown energy, the spatial variation throughout the rupture process considerably differs. Furthermore, the same spatial locations can have significantly different breakdown energy values in different rupture events of comparable size.

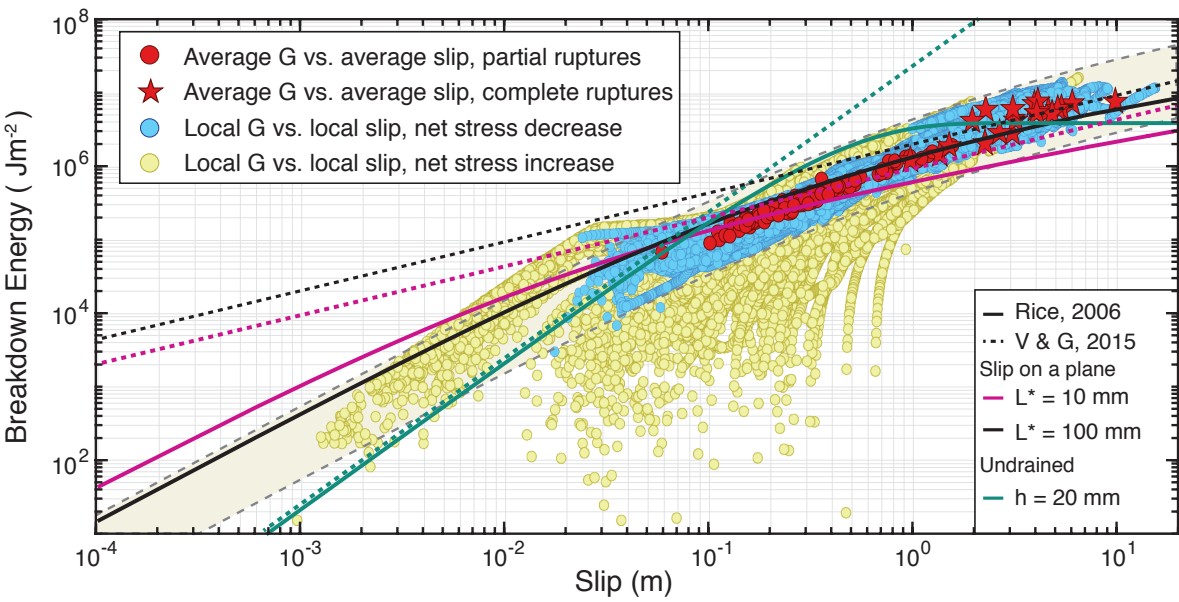

**Figure 11.** The average and local breakdown energy values for the simulated ruptures show an increasing trend with average and local slip, consistent with inferences from natural earthquakes (Fig. 4). The general trend of increasing breakdown energy with slip qualitatively holds for local points within our simulated ruptures; however, there is considerable variability for individual values of $G$ at a given slip. For points that exhibit net weakening behavior in our simulated ruptures (blue circles), local values of $G$ tend to vary within a factor of 3 from the scaling relationship between average $G$ and average slip. The shaded band bordered by grey dashed lines illustrates the variation in $G$ at a given value of slip. Local values of $G$ are more variable for regions that experience a net increase in stress during the rupture process (yellow circles), e.g., regions close to rupture arrest. Theoretical curves for $G$ vs. slip are indicated by solid lines for Eqs. (17) and (21) based on Rice (2006) and dashed lines for Eqs. (22 - 23) based on Viesca and Garagash (2015), with the coefficient of friction of $f = 0.53$ and values otherwise indicated in Table 1. In both cases, the magenta and black lines correspond to the solutions for slip on a plane with two different values of $L^*$ while the green line corresponds to the solution for an adiabatic and undrained shear band of width 20 mm.

*Author contributions.* V.L. and N.L. both contributed to developing the main ideas, designing the modeling, and producing the manuscript. V.L. carried out and analyzed the presented numerical experiments.

*Competing interests.* The authors declare no competing financial interests.

*Acknowledgements.* This study was supported by the National Science Foundation (grant EAR 1724686), the United States Geological Survey (grant G19AP00059), and the Southern California Earthquake Center (SCEC), contribution No. 19085. SCEC is funded by NSF Cooperative Agreement EAR-1033462 and USGS Cooperative Agreement G12AC20038. Numerical simulations for this study were carried out on the High Performance Computing Center cluster of the California Institute of Technology. We thank Eric Dunham and Elisa Tinti for helpful comments and suggestions that improved the manuscript.

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
