# Peer review of "Rupture-dependent breakdown energy in fault models with thermo-hydro-mechanical processes"

_Solid Earth, 2020_

## Short Comment (SC1) · 4 Aug 2020

We would like to kindly call the attention of the authors to some recent literature that appears to be directly relevant to their preprint.

The two recent papers: "The emergence of crack-like behavior of frictional rupture: The origin of stress drops", Physical Review X 9, 041043 (2019) [https://doi.org/10.1103/PhysRevX.9.041043] and "The emergence of crack-like behavior of frictional rupture: Edge singularity and energy balance", Earth and Planetary Science Letters 531, 115978 (2020) [https://doi.org/10.1016/j.epsl.2019.115978] extensively discuss the analogy between frictional rupture and ideal cracks, which is also

central to the authors' preprint. In particular, the latter paper extensively discusses the relations between the breakdown energy and the edge-localized fracture energy (and the associated length scales). It is found, for generic rate-and-state constitutive relations, that (i) part of the breakdown energy can be identified with a well-defined edge-localized fracture energy, which depends on the constitutive relation, that is balanced by the elastic energy flux associated with the edge singularity and that determines the rupture speed. (ii) The breakdown energy can significantly exceed the edge-localized fracture energy, a deviation that is associated with the intrinsic rate-dependence of friction and the lack of strict length scales separation (iii) The breakdown energy is position dependent, even for the very same earthquake rupture propagating along a spatially homogeneous fault.

These findings seem to be directly relevant for the authors' preprint.

We hope the authors find these comments useful for improving their preprint.

All the very best,

Prof. Eran Bouchbinder, Weizmann Institute of Science, Israel

Prof. Jean-François Molinari, École Polytechnique Fédérale de Lausanne, Switzerland

Dr. Efim Brener, Forschungszentrum Jülich, Germany

---

## Referee Comment (RC1) · E.M. Dunham (Referee) · 8 Aug 2020

This study examines the dependence of dissipated energy on rupture history using simulations of earthquake ruptures with thermal pressurization as the dominant fault weakening mechanism. Dissipated energy can be divided into work done by sliding against the residual strength of the fault (referred to in this study as dynamic strength) and breakdown energy. Breakdown energy is one of the few earthquake source properties that can be indirectly estimated from far-field seismic radiation, and numerous observational studies have explored the dependence of breakdown energy on earthquake magnitude, local slip, etc. Likewise, theoretical studies of proposed fault weak-

ening processes, like thermal pressurization, provide predictions of how breakdown energy depends on slip and various parameters (like thermal and fluid transport properties of the fault zone). However, in order to obtain closed form analytical solutions, these theoretical studies often make assumptions like constant slip velocity, that are unlikely to be to be met in reality. The current study utilizes more complex earthquake simulations with thermal pressurization that provide more realistic rupture and slip histories. The authors calculate breakdown energy, both locally at each point on the fault and in a suitably averaged sense, and compare to both theoretical predictions and observational constraints. The main conclusions are that local breakdown energy can exhibit large spatial variations across the fault, due the complex rupture history, and can be quite different from the average breakdown energy that is estimated from far-field seismic observations. The study thus provides an important caveat for researchers who hope to infer fault weakening mechanisms from seismic observations. The study is well designed and the manuscript is clearly written; I recommend publication after addressing the following minor comments:

1. Line 45. It is stated that peak and dynamic strengths are expected to be material properties of a fault, but I disagree that this is how people usually think about it. It is more common to regard static and dynamic friction coefficients as material properties, recognizing that shear strength depends on both friction and effective normal stress. I think it is widely understood that the ambient effective normal stress is not a material property of fault, but depends on tectonic loading and fluid state. I recommend explaining this in more detail, pointing out that for the set-up in your 2D simulations, the ambient effective normal stress is a prescribed quantity that is unaltered by fault slip (unlike in a dipping fault configuration, etc.).

2. Line 53. The LEFM relationship between rupture velocity and breakdown energy requires that the small-scale yielding criterion is met. You later explain this, but it might be helpful to mention small-scale yielding here. (Optional)

3. Line 60. I think you mean greater than 1 m/s, not 10ˆ3 m/s!

4. Line 136. "or" should be "of"

5. Equation (11). Should the integral be over Sigma, not Omega? In any case, please make sure to define Sigma and/or Omega.

6. Line 194, lines 258-260, line 339, and elsewhere. (Depending on how you choose to respond to this suggestion, this could warrant a major revision.) You compare your simulation results to the two closed-form thermal pressurization solutions in Rice (2006), both of which utilize the constant slip velocity assumption. However, this was improved upon by Viesca and Garagash (2015) to account for a more realistic slip velocity history that accounts, in the context of a steadily propagating rupture, for elastodynamic relations between slip and stress change. Viesca and Garagash along provide solutions for thermal pressurization and the dependence of breakdown energy on slip. I think your paper would be substantially strengthened by comparing your simulation results to their theoretical predictions, in addition to the Rice (2006) predictions. This might provide insight into the validity of a steady state solution for describing more complex ruptures that accelerate, decelerate, interact with arrest waves, etc. Perhaps there are situations where the steady state solution is a good approximation, or maybe not. It would be very useful to know this since it will help guide the field to either invest more time in developing steady state solutions for other weakening mechanisms or to instead shift toward fully numerical rupture simulations like you have done.

7. Figure 1a. What is the small white triangle? Should this region be shaded blue?

8. Figure 3 and elsewhere. You utilize a 2D model, but then make a comparison to observationally interred breakdown energy from real earthquakes in 3D. It would be useful to add a few sentences or a paragraph discussing whether or not the 2D idealization alters the predicted scaling behavior. Many people familiar with wave propagation understand that there are substantial differences between 2D and 3D, but for various reasons (discussed, for example, in Freund's Dynamic Fracture Mechanics textbook) this is far less the case for fracture problems. Please comment on this to avoid confusion

and to give readers more confidence that the comparison you've made is relevant.

9. Figure 5. It appears that there is weakening that is confined to very small slip, prior to the main effects of thermal pressurization. Is this due to the drop in friction coefficient from standard rate-and-state effects? If so, it might be possible to capture this (small) contribution to breakdown energy through a typical LEFM fracture energy idealization, as was done by Brantut and Rice (GRL, 2011). Consider commenting on this.

–Eric M. Dunham

---

## Referee Comment (RC2) · Elisa Tinti (Referee) · 25 Aug 2020

This paper describes earthquakes rupture histories inferred with dynamic simulations in which thermal pressurization (TP) has a dominant effect. On their dynamic simulations the authors study in particular the breakdown energy considering punctual estimates as well as global/average values.

One of the conclusions is that local breakdown energy can exhibit large spatial variations across the fault and large temporal variations on the same location in different earthquake ruptures.

[Figure]

In the literature, dynamic models of real events show heterogeneous distribution of dynamic parameters (usually in terms of initial stress) to reproduce seismological data.

I completely agree with the main goal of this paper because the authors try to explain an observed feature (the heterogeneity of dynamic models in space and in time) of real events.

The authors compare their theoretical results with estimates coming from real events and find many differences in the scaling relation. I think the study is well designed and the manuscript is clearly written however I have many moderate comments.

My first doubt concerns the definition of breakdown energy. In the literature there is still a confusion about the meaning of fracture energy and I think the authors must do an additional effort to clarify the meaning of this parameter. The shear cracks with the cohesive-zone have been proposed in the literature to overcome the singularity of early crack models and in these model fracture energy is considered as the energy absorbed behind the tip and needed to allow the crack to propagate. In these models, fracture energy surely contains the contribution of surface energy but not heat. In fact, all the dissipations are ascribed to frictional heating (the area below the minimum stress). Recently in the literature a new definition of fracture energy (as well as the discussion about the meaning of fracture energy on real fault planes) has been proposed to reconcile seismological measurements, geological observations and laboratory experiments and to obtain a coherent understanding of the governing physical processes. Tinti et al (2005) proposed to call "breakdown work" the area below the traction versus slip curve and above the minimum traction value reached when the slip is still increasing (it has been called "work" even if it is an energy, so we can think to it as a breakdown energy). The idea to change the definition and also the symbol (Wb instead of G) was made because all the potential contributions and processes that occur at different length scales during propagation can contribute to this energy: heat, breakage of the asperities as well as comminution of the fault gouge, thermal pressurization, flash heating or other processes absorbed in the fault plane, virtually assumed with an infinitesimal thickness.

For its general definition, this work (or energy) is not constrained to be absorbed just behind the rupture front. The area corresponding to the breakdown work is the only area we are able to "measure" during real events because we cannot know the absolute value of initial stress (it is usually assumed a priori). Because we represent the fault plane as a mathematical virtual plane this area can contain different contribution of energy, also heat.

Looking figure 5 and 7 it seems to me that the authors are generalizing the computation of breakdown energy in the same way.

Dynamic models proposed in the literature for real events have to choose a particular data-set of dynamic parameters (e.g., tau_yield, tau_dyn and Dc or mu_s, mu_dyn, sigma_n for SW law, or a,b,L for R&S law) that fix in some way the breakdown energy of a specific event. The choice of SW law imposes the G value a priori and does not allow for multiple seismic cycles while the use of R&S law in multiple events allows for a local and temporal variation of G.

I think the study can improve by:

1) showing the same figures (5 or 7) for dynamic simulations without TP;

2) trying to better explain the meaning of the breakdown energy;

3) focusing the interpretation of the results not only on the TP effects but on the difficulties with real events and with current resolution of data to constrain the slip velocity function. More info on this issue are surely useful also for kinematic modelers.

Specific comments

1) Line 17: shear crack or shear pulse (Heaton 1990).

2) Line 28: Probably in this sentence the authors would have mentioned Cocco and Tinti 2008 instead of Cocco et al 2004.

3) Figure 1: the authors should explicitly underline that the radiated energy can be

computed from the blue area only if the plot represents an average estimate of slip and stress and it is not a punctual plot – as the authors have correctly written in the axes labels (see Kanamori and Rivera 2006).

4) Line 45 : "If slip weakening were the fundamental constitutive behavior describing fault resistance during dynamic rupture, then its parameters - the peak resistance, dynamic resistance, and breakdown energy G - would be expected to be material properties". I think this sentence is not correct. The authors should say that in literature the SW law is frequently adopted for a single event because it allows to assume (or retrieve) only three parameters (mu_static and mu_dynamic and Dc) and because the slip-weakening behavior has been observed also with R&S law. Often, assuming a SW law simplifies the modeling but it does not mean that the authors believe that G is a material property as well as Dc. From laboratory it has been seen that the first two parameters (mu_static and mu_dynamic) are material properties even if they also can slightly change due to different conditions of the rock fabric and fault deformation (strain). Differently, Dc is a debated parameter because it is still not well constrained. So, make the inference that in the literature G is expected to be a material property is too strong and erroneous.

5) Line 52: "...and remaining variations in rupture speed are largely controlled by the breakdown energy in such linear slip-weakening representations (Guatteri and Spudich, 2000)." I don't understand this sentence.

6) Line 55: Many other papers suggest the relation among Breakdown energy and slip (Cocco and Tinti 2008, Brantut and Viesca 2017, Nielsen et al 2016, Selvadurai 2019)

7) Line 56: "which is inconsistent with the breakdown energy being a material property as assumed in linear slip-weakening laws". I don't understand: who believes that energy is a property of the materials?

8) Line 57: I suggest to add that Perry et al 2020 results have been obtained assuming constant L parameter on the fault plane.

9) Line 61: Nielsen et al 2016 say that their measures saturate in the rotary lab machine.

10) Line 65: Please cite Brantut and Viesca 2017.

11) Line 85: Also inferring the dynamic parameters from pseudo-dynamic models (Ide and Takeo 1997, Bouchon et al 1998, Tinti et al 2005, Causse et al 2014) suggest more complex slip weakening behaviors with heterogeneous traction evolutions and heterogeneous dynamic levels.

12) Line 88: I don't think the reader can understand the meaning of "Furthermore, the shear heating itself would depend not on the energy counted as "breakdown" but on the overall dissipated energy, making the fault weakening - and hence rupture dynamics - dependent on the absolute stress levels, and not just on stress changes, as typically considered by analogy with traditional fracture mechanics."

13) Why in figure 3 the authors didn't use a more complete dataset?

14) Lines 95-98: I perfectly agree with this sentence but I would further stress the issue of the slip rate because it's a significant uncertainty of kinematic models.

15) Equation (13) is essentially equivalent to equation (1) in Tinti et al 2005 (assuming only one component) or equation (7) in Cocco and Tinti 2008.

16) Line 182: I cannot appreciate the meaning because I cannot read Lambert et al in review. But I suppose that also in that case it depends on the assumption of minimum stress level to compute G during the slip evolution.

17) I suggest to change the symbol given to the breakdown energy because it is different to the meaning of fracture energy coming from cohesive-shear cracks.

18) How the conditions obtained in a 2D models to arrest the rupture can influence the results respect to a 3D model? How the 2D results have to be scaled to 3D to be compared with estimates from the literature?

19) Line 225: The authors should underline that the initial stress is the only hetero-geneous distribution included in these models and therefore it is the most important parameter that affects the size of the event.

20) Figure 7: I would expect a temporal variability of G for different events because fault properties vary with time (and the equivalent tau_yield and tau_min are varying during the seismic cycles). The observables up to date seem to give robust information of the average behavior of G but not on the local estimates. The main problem is that we are not able to infer the slip rate on the fault planes with the actual resolution of seismic data. Results shown in Figure 7 can stress the idea that is very difficult to constrain the local distribution of G because it can change frequently with time as demonstrated theoretically.

21) Figure 6-7: I really like these figures that show how different can be locally the traction evolution as a function of slip due to different slip velocities. I image that the different points on the fault show a very different traction evolution. I suspect that some-time is very difficult to decide which is the minimum traction to fix the area below the curve. Probably there exist many other fault points with heterogeneous slip weakening behavior whose dynamic values reached toward the final slip vary but lies above the first important minimum value, so the area doesn't not increase too much. Moreover, I expect to observe a similar behavior imposing local stress heterogeneities able to produce secondary cracks propagations. I have two comments for these figures: (1) I suggest to the authors to discuss about the importance of the knowledge of slip rate more in general and not only linked to TP. (2) in figure 7 the authors have selected particular points in which the slip rates have surely two peaks values due to the com-plexity of the rupture front (as I can see from panels A). Should the authors add a panel with the slip rate function? Central column represents a test case in which G is more similar because the rupture front is smoother and slip rate is simpler. This is the reason why in the literature the average estimates of G are considered more robust than local estimates, both when proposed by spontaneous dynamic models and when calculated

on pseudo dynamic models, i.e. constrained by the kinematics of the event.

22) Probably the reader needs to start from a simpler condition: How is Figure 7 if the authors simulate R&S law with homogeneous parameters, heterogeneous initial stress and without TP?

23) How is the traction evolution (figure 7) if the authors model an event with R&S law, homogeneous constitutive parameters and constant initial stress with TP (maybe it can be the first modeled event of the seismic cycle)? In this way the reader can appreciate when the effects of TP occur.

24) I suggest to the authors to write that the breakdown energy is the only measurable energy and a future challenge is to understand what does really it represent.

Technical corrections: 1) Line 60: high slip rate (>10^3 m/s) I think there is a mistake or in the number or in the units . . .This number is too high if represents m/s.

Literature to add in the references among many other papers:

1) Gu and Wong 1991

2) Tinti et al 2005. JGR

3) Causse et al 2014 GJI

4) Nielsen et al 2016: G: Fracture energy, friction and dissipation in earthquakes, J. Seismol,

5) Cocco et al 2016, On the scale dependence of earthquake stress drop, J. Seismol.

6) Selvadurai, P. A. (2019). JGR

7) Brantut and Viesca 2017

8) Bizzarri 2010, JGR

---

## Editor Comment (EC1) · Jianye Chen (Editor) · 4 Sep 2020

The article has been reviewed by two referees who both support the modelling effort in this study. Both underline that the study should be published after minor or moderate modifications. The clarity of the definition of fracture (breakdown) energy could be readily improved, in addition to the comparison with the quasi-steady-state solution given by previous studies. All the TP modelling work (as in the present study) use constant TP coefficeint, which makes the pressurization so efficient that tends to cause a nearly-complete stress drop (at least in the spring-slider model). I am wondering what could be the possible damping factor(s) for the TP process (e.g., a smaller TP

effect with decreaseing effective normal stress). Would these factors affect the fracture energy and the rupture process? This would need extra work, but it will be useful if the authors could at least comment on this.

---

## Author Comment (AC1) · 5 Oct 2020

Thank you for informing us about this work. The relatively modest variations of breakdown energy in models with standard rate-and-state friction have been examined in several studies already cited in our manuscript (Cocco & Bizzari, 2002; Rubin & Ampuero, 2005; Ampuero & Rubin, 2008; Lapusta & Liu, 2009; Perry et al., 2020). The suggested work on "The emergence of crack-like behavior of frictional rupture: Edge singularity and energy balance" provides additional insight and hence we added this citation.

---

## Author Comment (AC2) · 5 Oct 2020

**Referee 1 (E. M. Dunham)**
This study examines the dependence of dissipated energy on rupture history using simulations of earthquake ruptures with thermal pressurization as the dominant fault weakening mechanism. Dissipated energy can be divided into work done by sliding against the residual strength of the fault (referred to in this study as dynamic strength) and breakdown energy. Breakdown energy is one of the few earthquake source properties that can be indirectly estimated from far-field seismic radiation, and numerous observational studies have explored the dependence of breakdown energy on earthquake magnitude, local slip, etc. Likewise, theoretical studies of proposed fault weakening processes, like thermal pressurization, provide predictions of how breakdown energy depends on slip and various parameters (like thermal and fluid transport properties of the fault zone). However, in order to obtain closed form analytical solutions, these theoretical studies often make assumptions like constant slip velocity, that are unlikely to be to be met in reality. The current study utilizes more complex earthquake simulations with thermal pressurization that provide more realistic rupture and slip histories. The authors calculate breakdown energy, both locally at each point on the fault and in a suitably averaged sense, and compare to both theoretical predictions and observational constraints.

The main conclusions are that local breakdown energy can exhibit large spatial variations across the fault, due the complex rupture history, and can be quite different from the average breakdown energy that is estimated from far-field seismic observations. The study thus provides an important caveat for researchers who hope to infer fault weakening mechanisms from seismic observations. The study is well designed and the manuscript is clearly written; I recommend publication after addressing the following minor comments:

Thank you for the positive assessment of our work and comments that helped us improve the manuscript. Please find our responses to your comments below.

1. Line 45. It is stated that peak and dynamic strengths are expected to be material properties of a fault, but I disagree that this is how people usually think about it. It is more common to regard static and dynamic friction coefficients as material properties, recognizing that shear strength depends on both friction and effective normal stress. I think it is widely understood that the ambient effective normal stress is not a material property of fault, but depends on tectonic loading and fluid state. I recommend explaining this in more detail, pointing out that for the set-up in your 2D simulations, the ambient effective normal stress is a prescribed quantity that is unaltered by fault slip (unlike in a dipping fault configuration, etc.).

We agree that the statement is imprecise and it is not essential to the manuscript so we have removed it. We have added the discussion about potential variations in ambient effective normal stress.

2. Line 53. The LEFM relationship between rupture velocity and breakdown energy requires that the small-scale yielding criterion is met. You later explain this, but it might be helpful to mention small-scale yielding here. (Optional)

We have added this specific example where small-scale yielding is introduced on line 84 (now 98).

3. Line 60. I think you mean greater than 1 m/s, not 10ˆ3 m/s!

We have corrected the text with the correct value $10^{-3}$ m/s or 1 mm/s, which generally represents the initiation of enhanced weakening in the lab. Thank you for noting this typographical error.

4. Line 136. "or" should be "of"

Thank you.

5. Equation (11). Should the integral be over Sigma, not Omega? In any case, please make sure to define Sigma and/or Omega.

We have updated the notation to consistently use Omega to denote the ruptured domain and clarified this in the text.

6. Line 194, lines 258-260, line 339, and elsewhere. (Depending on how you choose to respond to this suggestion, this could warrant a major revision.) You compare your simulation results to the two closed-form thermal pressurization solutions in Rice (2006), both of which utilize the constant slip velocity assumption. However, this was improved upon by Viesca and Garagash (2015) to account for a more realistic slip velocity history that accounts, in the context of a steadily propagating rupture, for elastodynamic relations between slip and stress change. Viesca and Garagash along provide solutions for thermal pressurization and the dependence of breakdown energy on slip. I think your paper would be substantially strengthened by comparing your simulation results to their theoretical predictions, in addition to the Rice (2006) predictions. This might provide insight into the validity of a steady state solution for describing more complex ruptures that accelerate, decelerate, interact with arrest waves, etc. Perhaps there are situations where the steady state solution is a good approximation, or maybe not. It would be very useful to know this since it will help guide the field to either invest more time in developing steady state solutions for other weakening mechanisms or to instead shift toward fully numerical rupture simulations like you have done.

This is an excellent suggestion that we have implemented in the manuscript. The approximation of a steadily propagating rupture does not capture the variability in local G vs. slip as observed in our simulated dynamic ruptures. We have added the end-member curves for the drained and undrained cases from Viesca and Garagash (2015) to Fig. 9 (new Fig. 11), which exhibit marginal differences on the log-log plot from the solutions of Rice (2006). Both sets of solutions from Viesca & Garagash (2015) and Rice (2006) provide qualitative insight into the increase in G with slip, but do not provide much detail in the variability for individual points throughout ruptures.

7. Figure 1a. What is the small white triangle? Should this region be shaded blue?

The white section of the dashed red trapezoid in Figure 1a corresponds to the portion of the strain energy change per unit area that corresponds to the breakdown energy outside of the red trapezoid (that arises when the initial and peak shear stress do not coincide). This additional dissipated energy outside of the red trapezoid comes at the expense of the radiated energy. We have added this comment to the caption.

8. Figure 3 and elsewhere. You utilize a 2D model, but then make a comparison to observationally interred breakdown energy from real earthquakes in 3D. It would be useful to add a few sentences or a paragraph discussing whether or not the 2D idealization alters the predicted scaling behavior. Many people familiar with wave propagation understand that there are substantial differences between 2D and 3D, but for various reasons (discussed, for example, in Freund's Dynamic Fracture Mechanics textbook) this is far less the case for fracture problems. Please comment on this to avoid confusion and to give readers more confidence that the comparison you've made is relevant.

We have added a paragraph in section 5 discussing that the exact scaling relationship between breakdown energy and slip should be examined in 3D simulations. However, the main conclusions of this work that breakdown energy is a rupture-dependent quantity should be the same in 3D models.

9. Figure 5. It appears that there is weakening that is confined to very small slip, prior to the main effects of thermal pressurization. Is this due to the drop in friction coefficient from standard rate-and-state effects? If so, it might be possible to capture this (small) contribution to breakdown energy through a typical LEFM fracture energy idealization, as was done by Brantut and Rice (GRL, 2011). Consider commenting on this.

The initial weakening at very small slip in Fig. 5 (new Fig. 6) indeed mostly comes from the drop in friction coefficient due to the standard rate-and-state friction. The dynamic resistance level for the rateand-state component at 1-10 m/s slip rate would be tau_{ss}(V) = 13.3-13.0 MPa, comparable to where the rapid drop transitions to more gradual effect due to thermal pressurization, although, for some points, rapid weakening due to adiabatic thermal pressurization seems to start within that small slip as well. It would be difficult to accurately estimate the breakdown energy involved apriori, as we know only the slope of that weakening - b*sigma/D_rs - but not the peak stress (which depends on the pre-rupture state variable), the very small slips involved, or how much thermal pressurization is mixed in. However, assuming that the thermal pressurization is not yet involved, and taking typical inter-event times 10 years as an estimate for the pre-rupture state variable, we can get the upper bound on the breakdown energy associated with the rate-and-state processes to be 0.15 MJ/m$^2$, much smaller than the overall breakdown energies we obtain. We have added a comment on this to the text.

---

## Author Comment (AC3) · 5 Oct 2020

Referee 2 (E. Tinti)
This paper describes earthquakes rupture histories inferred with dynamic simulations in which thermal pressurization (TP) has a dominant effect. On their dynamic simulations the authors study in particular the breakdown energy considering punctual estimates as well as global/average values. One of the conclusions is that local breakdown energy can exhibit large spatial variations across the fault and large temporal variations on the same location in different earthquake ruptures.

In the literature, dynamic models of real events show heterogeneous distribution of dynamic parameters (usually in terms of initial stress) to reproduce seismological data. I completely agree with the main goal of this paper because the authors try to explain an observed feature (the heterogeneity of dynamic models in space and in time) of real events. The authors compare their theoretical results with estimates coming from real events and find many differences in the scaling relation. I think the study is well designed and the manuscript is clearly written however I have many moderate comments.

Thank you for the positive assessment of our work and comments that helped us improve the manuscript. Please find our responses to your comments below.

My first doubt concerns the definition of breakdown energy. In the literature there is still a confusion about the meaning of fracture energy and I think the authors must do an additional effort to clarify the meaning of this parameter. The shear cracks with the cohesive-zone have been proposed in the literature to overcome the singularity of early crack models and in these model fracture energy is considered as the energy absorbed behind the tip and needed to allow the crack to propagate. In these models, fracture energy surely contains the contribution of surface energy but not heat. In fact, all the dissipations are ascribed to frictional heating (the area below the minimum stress).

Recently in the literature a new definition of fracture energy (as well as the discussion about the meaning of fracture energy on real fault planes) has been proposed to reconcile seismological measurements, geological observations and laboratory experiments and to obtain a coherent understanding of the governing physical processes. Tinti et al (2005) proposed to call "breakdown work" the area below the traction versus slip curve and above the minimum traction value reached when the slip is still increasing (it has been called "work" even if it is an energy, so we can think to it as a breakdown energy). The idea to change the definition and also the symbol (Wb instead of G) was made because all the potential contributions and processes that occur at different length scales during propagation can contribute to this energy: heat, breakage of the asperities as well as comminution of the fault gouge, thermal pressurization, flash heating or other processes absorbed in the fault plane, virtually assumed with an infinitesimal thickness.

For its general definition, this work (or energy) is not constrained to be absorbed just behind the rupture front. The area corresponding to the breakdown work is the only area we are able to "measure" during real events because we cannot know the absolute value of initial stress (it is usually assumed a priori). Because we represent the fault plane as a mathematical virtual plane this area can contain different contribution of energy, also heat.

Looking figure 5 and 7 it seems to me that the authors are generalizing the computation of breakdown energy in the same way. Dynamic models proposed in the literature for real events have to choose a particular data-set of dynamic parameters (e.g., tau_yield, tau_dyn and Dc or mu_s, mu_dyn, sigma_n for SW law, or a,b,L for R&S law) that fix in some way the breakdown energy of a specific event. The choice of SW law imposes the G value a priori and does not allow for multiple seismic cycles while the use of R&S law in multiple events allows for a local and temporal variation of G. I think the study can improve by:
1) showing the same figures (5 or 7) for dynamic simulations without TP;
2) trying to better explain the meaning of the breakdown energy;
3) focusing the interpretation of the results not only on the TP effects but on the difficulties with real events and with current resolution of data to constrain the slip velocity

function. More info on this issue are surely useful also for kinematic modelers.

The definition of breakdown energy/work in this study is indeed consistent with that of Tinti et al. (2005). In solid mechanics and engineering, the cohesive-zone models of fracture mechanics have been extensively applied to cracks with all sorts of inelastic processes around the crack tip since 1960s, as long as the inelastic zone is small (small-yield assumption mentioned in the manuscript). In those cases, the fracture energy G has already been routinely used to refer to inelastic dissipation broadly, with damage, plastic work, frictional heat etc. often dominating the surface energy. Consider the publication of J. R. Rice, "The Mechanics of Earthquake Rupture", in Physics of the Earth's Interior (Proc. International School of Physics 'Enrico Fermi', Course 78, 1979; ed. A. M. Dziewonski and E. Boschi), Italian Physical Society and North-Holland Publ. Co., 1980, pp. 555-649. It stated: "mathematically, G is the rate (with respect to crack area) of energy loss through the singularity, and physically it is the energy flow to "breakdown" processes at the tip, i.e. the fracture energy" (p. 586), and "microscale cracking processes involved in macroscopic shear faulting will be complex, and will have associated with them a far greater effective fracture energy than for a single tensile crack" (p. 589). The textbook on "Dynamic fracture mechanics" by L. B. Freund uses an example of a cohesive zone calculation based on a (small) plastic zone in front of the crack tip; in that case, physically, most of the associated plastic work that goes into "fracture energy" is actually heat.

We have added a discussion about breakdown energy vs. fracture energy to the introduction.

Specific comments
1) Line 17: shear crack or shear pulse (Heaton 1990).

Thank you for the suggestion, we have added it to the text.

2) Line 28: Probably in this sentence the authors would have mentioned Cocco and Tinti 2008 instead of Cocco et al 2004.

We have updated the reference to Tinti et al. (2005) which appears to the most appropriate reference dedicated to this topic.

2) Figure 1: the authors should explicitly underline that the radiated energy can be computed from the blue area only if the plot represents an average estimate of slip and stress and it is not a punctual plot – as the authors have correctly written in the axes labels (see Kanamori and Rivera 2006).

We have added text emphasizing that the energy balance in Fig. 1 holds for the average stress vs. slip curve for the entire source process, and not the local behavior.

Note that relationship between the energy partitioning, including the radiated energy, and the average stress-slip curve depends on the particular construction of the curve by the averaging of the local evolution stress and slip as discussed in Noda & Lapusta (2012). This is already discussed in section 3 but we have added a more explicit reference to it.

4) Line 45 : "If slip weakening were the fundamental constitutive behavior describing fault resistance during dynamic rupture, then its parameters - the peak resistance, dynamic resistance, and breakdown energy G - would be expected to be material properties".

I think this sentence is not correct. The authors should say that in literature the SW law is frequently adopted for a single event because it allows to assume (or retrieve) only three parameters (mu_static and mu_dynamic and Dc) and because the slip-weakening behavior has been observed also with R&S law. Often, assuming a SW law simplifies the modeling but it does not mean that the authors believe that G is a material property as well as Dc. From laboratory it has been seen that the first two parameters (mu_static and mu_dynamic) are material properties even if they also can slightly change

due to different conditions of the rock fabric and fault deformation (strain). Differently, Dc is a debated parameter because it is still not well constrained. So, make the inference that in the literature G is expected to be a material property is too strong and erroneous.

We agree and we have removed this sentence. The rest of the paragraph in the manuscript already follows the suggested discussion.

We have added text and a figure (Fig. 3) that demonstrate that f_peak, f_dyn, and D_c are also not fixed quantities for the standard rate-and state friction.

5) Line 52: ": : :and remaining variations in rupture speed are largely controlled by the breakdown energy in such linear slip-weakening representations (Guatteri and Spudich, 2000)." I don't understand this sentence.

In the direct analogy between breakdown energy and fracture energy for linear slip-weakening friction, the balance between breakdown energy and the released strain energy would be expected to govern the rupture speed. As mentioned in Guatteri and Spudich (2000), the distribution of stress drop and breakdown energy are assumed by seismologists to be the best constrained parameters, under standard slip-weakening descriptions of friction. Static stress drop can be inferred from the final slip distribution and is related to the strain energy released (assuming standard fracture mechanics and ignoring undershoot/overshoot), which allows the breakdown energy to be inferred from variations in rupture speed.

6) Line 55: Many other papers suggest the relation among Breakdown energy and slip (Cocco and Tinti 2008, Brantut and Viesca 2017, Nielsen et al 2016, Selvadurai 2019)

We have added these references as additional evidence for the scaling relationship.

7) Line 56: "which is inconsistent with the breakdown energy being a material property as assumed in linear slip-weakening laws". I don't understand: who believes that energy is a property of the materials?

We have altered the term "material property" to "fixed fault property," which is what we meant to say, and added an explanation in the text. Many in the seismological community have this belief and base their studies on it. For example, some modeling studies have considered faults with assigned heterogeneous Dc and hence Gc values, treating them as properties of the interface, and considered sequences of events over such interfaces (e.g., Ide and Aochi, JGR, 2005; Aochi and Ide, 2011). It is also often implicit in ground motion studies that if one can determine the prestress and breakdown energy over the fault, then one may be able to predict the future rupture behavior, as would be the case in a classical linear elastic fracture problem, as already discussed in the conclusions.

8) Line 57: I suggest to add that Perry et al 2020 results have been obtained assuming constant L parameter on the fault plane.

We have added this in the text.

9) Line 61: Nielsen et al 2016 say that their measures saturate in the rotary lab machine.

We have added text mentioning that qualitatively similar scaling relationships for G and slip are observed in the lab and that saturation observed by Nielsen et al. (2016).

10) Line 65: Please cite Brantut and Viesca 2017.

We have added this reference in the text.

**11) Line 85: Also inferring the dynamic parameters from pseudo-dynamic models (Ide and Takeo 1997, Bouchon et al 1998, Tinti et al 2005, Causse et al 2014) suggest more complex slip weakening behaviors with heterogeneous traction evolutions and heterogeneous dynamic levels.**

We have added text mentioning this.

**12) Line 88: I don't think the reader can understand the meaning of "Furthermore, the shear heating itself would depend not on the energy counted as "breakdown" but on the overall dissipated energy, making the fault weakening - and hence rupture dynamics - dependent on the absolute stress levels, and not just on stress changes, as typically considered by analogy with traditional fracture mechanics."**

We have rephrased this sentence to improve clarity.

**13) Why in figure 3 the authors didn't use a more complete dataset?**

The values of average simulated breakdown energy, which are compared to the seismological inferences, are from the complete data set for the simulation. In Fig. 4 (old Fig. 3), we include the values of local breakdown energy for locations that experiences a net stress decreases, and hence a clear breakdown of shear stress. This is to illustrate that the continued weakening is thought to translate into both an increase in average and local breakdown energy with increased slip. This is the same simulated data presented in Fig. 11 (old Fig. 9), where we include the points that exhibit a net increase in shear stress (yellow).

As now clarified in the text (section 5), these additional points in Fig 11 (old Fig. 9, in yellow) highlight further conceptual complications in the partitioning of dissipated energy.

**14) Lines 95-98: I perfectly agree with this sentence but I would further stress the issue of the slip rate because it's a significant uncertainty of kinematic models.**

We have added further illustration and discussion on the variability of slip rate throughout ruptures in the text (e.g. new Fig. 8).

**15) Equation (13) is essentially equivalent to equation (1) in Tinti et al 2005 (assuming only one component) or equation (7) in Cocco and Tinti 2008.**

This general formulation for the energy associated with the breakdown process was introduced by Palmer & Rice (1973) and Rice (1980) and they have been already cited; we have added the citation to Tinti et al. (2005) as the more recent reference with the same focus. We have added reference to Cocco and Tinti. (2008) in a different portion of the manuscript, with regards to the clarification of breakdown work/energy vs fracture energy.

**16) Line 182: I cannot appreciate the meaning because I cannot read Lambert et al in review. But I suppose that also in that case it depends on the assumption of minimum stress level to compute G during the slip evolution.**

Yes, Lambert et al. (in review) explore estimates of average breakdown energy based on dynamic fracture theory for ruptures that exhibit a substantial stress undershoot, such as self-healing pulses. We find that the standard energy balance from the idealized energy diagram (Fig. 1), based on fracture mechanics, does not apply for such ruptures. The Lambert et al. (in review) should be published before this manuscript and we will provide a full citation here.

**17) I suggest to change the symbol given to the breakdown energy because it is different to the meaning of fracture energy coming from cohesive-shear cracks.**

We would prefer to keep the notation as is. As already discussed in response to an earlier comment, the fracture energy G of cohesive-shear models has been routinely understood to refer to all types of dissipated energy at the crack/rupture tip, including plastic work, damage, and frictional heat, since at least 1980s. There is an important value in using the notation of G since it is widely used. We think that it is especially important to continue to use the notation G as a link to numerous previous works in seismology and to make it widely known that G almost never relates to only surface energy as originally thought by Griffith (and erroneously repeated in selected geophysical publications), but rather represents the dissipation (and breakdown, which can also be called fracture) more broadly, even for tensile cracks.

18) How the conditions obtained in a 2D models to arrest the rupture can influence the results respect to a 3D model? How the 2D results have to be scaled to 3D to be compared with estimates from the literature?

We have added a paragraph in section 5 discussing that the exact scaling relationship between breakdown energy and slip should be examined in 3D dynamic rupture simulations, as the spatial distribution of slip and stress evolution differ between 2D and 3D in a nonlinear manner. However, the main conclusions of this work that breakdown energy is a rupture-dependent quantity should be the same in 3D models.

19) Line 225: The authors should underline that the initial stress is the only heterogeneous distribution included in these models and therefore it is the most important parameter that affects the size of the event.

We agree that the prestress before the rupture is the only heterogeneity, other than the VW-VS boundaries, as already discussed in the text. However, emphasizing only the prestress, without recognizing the role of the dynamic stress transfers and the weakening behavior for controlling the size of the rupture, is an oversimplification, as we discuss in the text (section 5). We have included a new figure (Fig. 8) to illustrate. The evolution of shear stress at individual points is due in part from the initial stress and local weakening, as well as the dynamic stress interactions during the rupture process which depend on the initial stress and weakening behavior elsewhere in the rupture. For example, the same prestress distribution would result in different rupture behavior for different efficiency of weakening (Fig. 8). Therefore, it is not enough to know the prestress without knowing the distribution of properties controlling the weakening behavior as well as the dynamic stress redistribution during and throughout the rupture.

20) Figure 7: I would expect a temporal variability of G for different events because fault properties vary with time (and the equivalent tau_yield and tau_min are varying during the seismic cycles). The observables up to date seem to give robust information of the average behavior of G but not on the local estimates. The main problem is that we are not able to infer the slip rate on the fault planes with the actual resolution of seismic data. Results shown in Figure 7 can stress the idea that is very difficult to constrain the local distribution of G because it can change frequently with time as demonstrated theoretically.

Indeed, this is the motivation for calling G a rupture-dependent quantity, as it varies at the same location over different ruptures. This is emphasized throughout the text and figures.

21) Figure 6-7: I really like these figures that show how different can be locally the traction evolution as a function of slip due to different slip velocities. I image that the different points on the fault show a very different traction evolution. I suspect that sometime is very difficult to decide which is the minimum traction to fix the area below the curve. Probably there exist many other fault points with heterogeneous slip weakening behavior whose dynamic values reached toward the final slip vary but lies above the first important minimum value, so the area doesn't not increase too much. Moreover, I expect to observe a similar behavior imposing local stress heterogeneities able to produce secondary cracks propagations. I have two comments for these figures:

(1) I suggest to the authors to discuss about the importance of the knowledge of slip rate more in general and not only linked to TP.
(2) in figure 7 the authors have selected particular points in which the slip rates have surely two peaks values due to the complexity of the rupture front (as I can see from panels A). Should the authors add a panel with the slip rate function? Central column represents a test case in which G is more similar because the rupture front is smoother and slip rate is simpler. This is the reason why in the literature the average estimates of G are considered more robust than local estimates, both when proposed by spontaneous dynamic models and when calculated on pseudo dynamic models, i.e. constrained by the kinematics of the event.

These are excellent suggestions. We have included a new figure (new Fig. 8) comparing ruptures produced for the same initial stress with and without TP. This demonstrates how TP facilitates rupture propagation under lower prestress, and how local slip rate and stress evolution are highly variable through ruptures. For the R&S friction, the difference in the stress evolution for breakdown is relatively mild, due to the logarithmic dependence on slip rate. This effect is substantially larger in ruptures with TP. Note that the slip rate and stress evolution does not only depend on local prestress but the prestress and weakening conditions throughout the entire rupture through dynamic stress interactions.

22) Probably the reader needs to start from a simpler condition: How is Figure 7 if the authors simulate R&S law with homogeneous parameters, heterogeneous initial stress and without TP?

We have included a new figure (Fig. 8) illustrating the evolution of shear resistance and slip rate in a rupture with only standard rate-and-state friction vs. with TP. We also have added a new figure (Fig. 3) which demonstrates how peak and dynamic friction, as well as slip-weakening distance depend on sliding rate. More detailed examination of the stress evolution and breakdown energy for rate-and-state friction have been shown in a number of previous work, which are referenced along with the new Fig. 3.

23) How is the traction evolution (figure 7) if the authors model an event with R&S law, homogeneous constitutive parameters and constant initial stress with TP (maybe it can be the first modeled event of the seismic cycle)? In this way the reader can appreciate when the effects of TP occur.

We have included a new figure illustrating this (Fig. 8).

24) I suggest to the authors to write that the breakdown energy is the only measurable energy and a future challenge is to understand what does really it represent.

We agree that the physical interpretation of breakdown energy is an open question and a future challenge, and we emphasize this in the conclusions. However, the breakdown energy is a derivative inference and not, as far as we know, a directly measurable quantity. The energy that we hope one can infer seismologically is the radiated energy. The average breakdown energy is then estimated assuming a specific rupture model and relationship between the inferred static stress drop, average slip, radiated energy, and average breakdown energy, as in formula (14) in the manuscript. It is an open question as to whether the local breakdown energy can be reliably inferred, since its relationship to rupture speed is not evident. It is possible that additional constraints can be placed on the total dissipated energy density from thermal arguments, but such constraints would be more of an upper bound, as they would be directly relevant to the specific portion attributed to the breakdown process if the representative residual shear resistance is near zero.

Technical corrections: 1) Line 60: high slip rate (>10^3 m/s) I think there is a mistake or in the number or in the units : : :This number is too high if represents m/s.

Yes, there was a typo, the value should be $10^{-3}$ m/s. We have corrected this in the text.

Literature to add in the references among many other papers:
1) Gu and Wong 1991
2) Tinti et al 2005. JGR
3) Causse et al 2014 GJI
4) Nielsen et al 2016: G: Fracture energy, friction and dissipation in earthquakes, J. Seismol,
5) Cocco et al 2016, On the scale dependence of earthquake stress drop, J. Seismol.
6) Selvadurai, P. A. (2019). JGR
7) Brantut and Viesca 2017
8) Bizzarri 2010, JGR

We have included most of these references in our manuscript.

---

## Author Comment (AC4) · 5 Oct 2020

The emphasis of this work is to demonstrate that even in fault models with constant and uniform TP parameters, the resulting breakdown energy is heterogeneous and not constant in time, thus it is rupture-dependent. Given suggestions from the reviewers, we now expand on this to discuss the importance of dynamic rupture simulations for evaluating the variability and physical interpretation of breakdown energy in fault models with thermo-hydro-mechanical processes, compared to steady-state rupture solutions.

The TP parameters used in this work represent moderate weakening motivated by our previous work and other prior studies (e.g. Rice, 2006), and do not result in complete

stress drop (as is seen Figures 5-10). These parameters have also been shown in our prior studies to be able to qualitatively and quantitatively reproduce a number of seismological observations, including magnitude invariant static stress drops between 1-10 MPa, the inferred scaling and values of breakdown energy from moderate to large earthquakes, and radiation efficiencies between 0.1 to 1 (Perry et al., 2020; Lambert et al., in review). These studies suggest that fault models with such TP parameters may be plausible representations of natural mature faults, at least megathrust faults.

Indeed, heterogeneous and non-constant hydraulic properties, as may result from damage generation during rupture, would further complicate the evolution of shear stress with thermo-hydro-mechanical processes like TP, as was already discussed in lines 312 – 317 of the original manuscript (now 115- 119 and 434 - 439). As damage generation is also likely to be rupture-dependent, this is also expected to reinforce our conclusions that breakdown energy is rupture-dependent, as discussed in the conclusion section.